

# Calibration of channel depth and friction parameters in the LISFLOOD-FP hydraulic model using medium resolution SAR data

M. Wood [1,2], R. Hostache [2], J. Neal [1], T. Wagener [3], L. Giustarini [2], M. Chini [2], G. Corato [2], P. Matgen [2] & P. Bates [1].

1. School of Geographical Sciences, University of Bristol, University Road, Bristol, BS8 1SS UK.
2. Luxembourg Institute of Science and Technology (LIST), 41, rue du Brill, L-4422 Belvaux, Luxembourg.
3. School of Engineering, University of Bristol, Queen's Building, University Walk, Bristol BS8 1TR.

## 1 Abstract

Single satellite Synthetic Aperture Radar (SAR) data are now regularly used to estimate hydraulic
model parameters such as channel roughness, depth and water slope. However despite channel
geometry being critical to the application of hydraulic models and poorly known *a priori*, it is not
frequently the object of calibration. This paper presents a unique method to calibrate
simultaneously the bankfull channel depth and channel roughness parameters within a 2D
LISFLOOD-FP hydraulic model using an archive of moderate (75m) resolution SAR satellite-derived
flood extent maps and a binary performance measure for a 30x50km domain covering the
confluence of the rivers Severn and Avon in the UK. The unknown channel parameters are located by
a novel technique utilising the Information Content and identifiability of single and combinations of
SAR flood extent maps to find the optimum images for model calibration. Highest Information
Content is found in those SAR flood maps acquired near to the peak of the flood hydrograph, and
improves when more images are combined. We found model sensitivity to variation in channel
depth is greater than for channel roughness and a successful calibration for depth could only be
obtained when channel roughness values were confined to a plausible range. The calibrated reach-
average channel depth was within 0.9m (16% error) of the equivalent value determined from river
cross section survey data, demonstrating that a series of moderate resolution SAR data can be used
to successfully calibrate the depth parameters of a 2D hydraulic model.

## 19 Introduction

Flooding of over one third of the world's land area affected more than 2 billion people - 38% of the
world's population – between 1985 and 2003 (Dilley *et al.*, 2005).  Climate change forecasts also
indicate that in the future there may be an increase in the frequency and pattern of flooding
(European Environment Agency, 2012, European Commission, 2014, IPPC, 2014). One response to
this global hazard has been an increasing demand for better flood forecasts (Schumann *et al.*,
2009a).  Flood inundation models have an important role in flood forecasting and there has been
scientific interest in combining direct observations of flooding from remote sources with these
inundation models to improve predictions because of the persistent decline in the number of
operational gauging stations (Biancamaria *et al.*, 2010), and the reality that many river basins are
inaccessible for ground measurement. Synthetic Aperture Radar (SAR) satellites have particular
importance in this respect as they can discriminate between land and smooth open water surfaces
over large scales. These microwave (radar) frequency satellites are capable of all-weather day/night
observations and this makes them a particularly attractive option for observing floods. Currently





active SAR satellites include RADARSAT-2, ALSOS-2/PALSAR-2, TerraSAR-X, TanDEM-X, Sentinel 1 and
the COSMO SkyMed constellation. Historic data are also available from SAR satellites now out of
operation such as ENVISAT, ERS1 and 2 and RADARSAT-1.
By processing SAR data it is possible to produce binary maps of flood extent that can then be used,
either on their own, or intersected with a Digital Elevation Model (DEM) to produce shoreline water
levels, for model calibration and validation. Integration of SAR data with models is an established
technique for reducing uncertainty in model predictions as it updates/calibrates the model
states/parameters with observed data (e.g. Andreadis *et al.*, 2007, Biancamaria *et al.*, 2011b,
Domeneghetti *et al.*, 2014, Giustarini *et al.*, 2011, Garcia-Pintado *et al.*, 2013 and 2015, Hostache *et*
*al.*, 2009, Matgen *et al.,* 2010, Mason *et al.*, 2009 and 2012, Tarpanelli *et al.,* 2013), with the aim of
improving flood forecasts. Naturally, calibration of these hydraulic models is essential for accurate
results, and calibration studies to date have largely focussed on roughness. Aronica *et al.* (2002),
Tarpanelli *et al.,* 2013 , Hall *et al.,* 2005 and Di Baldassarre *et al.* (2009a, 2010 & 2011) and others
have used flood extent maps to successfully find best fit roughness parameter values. Schumann *et*
*al.* (2007) state that identifiability of parameters is important in order to obtain acceptable model
results since roughness factors can vary with location and in time. Mason *et al.* (2003) point to
roughness being a dominant factor for shallow reaches in particular and Di Baldassarre *et al.* (2009b)
found that the optimal roughness parameters depend on the timing of the SAR image and the
magnitude of the flood event. Historic observations of flooding should therefore have a particular
role in model calibration and sensitivity testing.
Despite the focus on calibrating unknown roughness values, the provision of good bathymetric data
is also critical to the application of hydraulic models (Trigg *et al.*, 2009, Legleiter *et al.* 2009).
Generally there are few ways to obtain bathymetry information for hydraulic models where no
ground data measurements exist. River depth may be estimated (e.g. Durand *et al.,* 2010 employed
an algorithm based on the Manning equation) or measured with optical satellites using reflectance
as Legleiter *et al.* showed (though the method is best suited to clear and shallow streams). Hostache
*et al.*, (2015) also proposed a drifting GPS buoy to assimilate water elevation and slope data into a
hydraulic model to define riverbed bathymetry, but overall passive and remote mechanisms are
scarce. Spatially distributed river depths are rarely available and there is a strong argument that
where channel geometry is *a priori* unknown it should also be estimated through calibration. It has
commonly been thought that channel geometry and roughness traded off against each other (e.g. as
in the well-known Manning equation) and therefore that they could not be uniquely identified at the
same time.  However, Garcia-Pintado *et al.* (2015) estimated channel friction and spatially-variable
channel bathymetry together using water levels derived from a sequence of real SAR overpasses (3m
resolution data from the COSMO-SkyMed constellation of satellites) and the Ensemble Transform
Kalman Filter. Though relating more specifically to depth of flow, rather than depth of channel,
Durand *et al.* (2008) demonstrated that estimates of depth and water (i.e. friction) slope could be
derived simultaneously from synthetic observations of water surface elevation integrated with a
hydraulic model. Yoon et al. (2012) were also able to derive bed elevations from similar synthetic
data. Mersel *et al.* (2013) progressed this further by proposing a slope-break method to locate
optimal locations to measure flow depth, through low to high flows over time, using synthetic data.
Durand *et al.,* Yoon *et al.* and Mersel *et al.* used synthetic altimetry data which was created within
the context of the upcoming Surface Water & Ocean Topography (SWOT) mission that will be able to
resolve rivers over 100m wide only.



Research to date has therefore demonstrated the feasibility of calibrating hydraulic model parameters governing channel depth and channel roughness simultaneously, using the higher spectrum resolution (up to 50m resolution) SAR images of flood extent. But because pixel size is inversely proportional to orbit revisit time, high resolution data are available only infrequently. There is thus some benefit to exploring the use of existing moderate (50m to 300m) resolution SAR data (such as the archive of 150m resolution ENVISAT Wide Swath Mode) to understand more about how channel depth and friction can be identified concurrently using coarser resolution SARs, and whether a single SAR flood map is sufficient to achieve this or a sequence of flood maps are more beneficial.

This paper draws on this prior research for simultaneous channel roughness and depth calibration and extends it by incorporating the use of an identifiability technique presented by Wagener *et al.* (2003), namely Dynamic Identifiability Analysis (DYNIA). In using flood extent for calibration the methodology also incorporates the under and over prediction of a model in accuracy scoring and disregards the correct detection of 'no water' pixels, thus adding extra information to the evaluation process.

Consequently, the objective of this paper is to determine whether medium resolution SAR data can be used to concurrently estimate channel friction and geometry parameters in a hydraulic model. If so, to determine if a single SAR derived flood map is sufficient to do this, or if a sequence of flood maps is better. A secondary aim is to test the utility of the DYNIA identifiability technique in this specific context to find the SAR images with high parameter information and locate the likely optimum parameter values. In section 1 we describe the methodology with information on the hydraulic model, the data needed to run it and the methods used to select the range of model parameters. There is also an introduction to the procedure used to process the satellite data and create flood extent maps from the SAR data. Section 2 describes the study area and data used, whilst Section 3 presents and discusses the results (including whether SAR observations at particular times during a flood or particular combinations of images are more successful). Conclusions are presented in Section 4.

# 1  Method

## 1.1  Hydraulic model

We use the LISFLOOD-FP hydraulic model with the Sub-Grid formulation of Neal *et al.* (2012) to simulate flood flows. LISFLOOD-FP (Bates and De Roo, 2000) is a 2D hydraulic model for subcritical flow that solves the local inertial form of the shallow water equations using a finite difference method on a staggered grid. As input the model requires ground elevation data describing the floodplain topography, channel bathymetry information (river width, depth and shape), boundary condition data consisting of discharge time series at all inflow points to the domain, water surface elevation time series at all outflow points and friction parameters which typically distinguish different values for the channel and floodplain. Of these data floodplain topography information is readily available from airborne and satellite Digital Elevation Models, boundary condition data can be taken from ground gauges, hydrologic models or statistical distributions, and friction parameters are typically estimated from lookup tables or calibrated. Channel bathymetry can be taken from ground surveyed cross sections, however for much of the planet no such measurements exist and



are impossible to obtain remotely.  In this situation channel bathymetry is *a priori* unknown and it is
therefore sensible to also treat it as a parameter that must be calibrated along with the friction.
In order to describe bathymetry as a calibrated variable in this experiment, river channel depth was
parameterised as a linear scaling of reach-average width using a single parameter '*r*'. This very basic
scaling of width was chosen so that only one bathymetry parameter needed to be estimated. This
simple approach will not be appropriate over an entire river network where the reach-averaged
width to depth relationship would be expected to change with bankfull discharge. However, the
width of the river chosen as a test case for this paper is constant along the simulated reach, while we
assume the depth of tributaries has an insignificant impact on the flooding on the main stem. In
effect the optimisation problem therefore simplifies to estimating reach-averaged bankfull depth
and Manning's '$n_c$' for a channel of reach-average width.
In width-varying river systems a dual parameterisation approach for depth and width could be
adopted but would substantially complicate the parameter estimation problem. The floodplain
Manning's roughness coefficient was assumed constant in these experiments as previous tests have
shown that the model was less sensitive to floodplain friction than channel friction .
We used Latin Hypercube Sampling (LHS) to take 1000 samples of the two uncertain LISFLOOD-FP
parameters '*r*' and channel Manning's roughness '$n_c$'. LHS is a useful sampling scheme for multiple
variables as the method can sample parameter values within a prior distribution in more than one
dimension (Huntington, 1998). We used LHS here as it is an efficient scheme that statistically
represents the parameter space without repetitions (Beven, 2009).

## 1.2   SAR image processing algorithm

Because SAR satellites are capable of all-weather day and night observations and can distinguish the
differences between land and open water signal returns they are particularly useful for observations
of flooding. To derive flood extent maps from the SAR images, we adopted the method proposed by
Matgen *et al.* (2011) and developed by Giustarini *et al.* (2013).
This method has three steps. Firstly the probability density function (pdf) of the open water
backscatter values in the SAR data is estimated. This requires identification of the bimodal aspect to
a histogram of backscatter values so that 'open water' values can be recognized from other
backscatter values. A theoretical pdf of water backscatter is then fitted to this histogram using
nonlinear regression techniques. The backscatter value where this pdf starts to diverge from the
histogram is identified. Then isolating those pixels with backscatter values lower than this threshold
produces a preliminary flood map.
The second step is to apply a region growing approach to grow the flooded areas within the
preliminary flood map until a tolerance level is reached. For the SAR image this step refines the
extent of pixels with an open water value.
In the last step a reference image is used to remove pixels from the flood map that do not change
between the flood and non-flood images (Hostache *et al.,* 2012) – i.e. pixels which have 'water
surface like' radar responses and could be either bodies of permanent water or smooth surfaces
such as car parks or flat roofs. This third step creates the final binary map of flood extent. Errors
inherent in the SAR processing are, for simplicity, not considered in this paper.





### 1.3  Performance measures

We compare these SAR derived flood maps against the simulated flood maps generated from
LISFLOOD-FP output at the equivalent time step by using a contingency matrix shown in Table 1.
Flood maps are compared pixel to pixel to determine if there is agreement or disagreement between
the two paired maps on whether there is surface water present or not.
**Table 1 Contingency table (after Stephens et al, 2014 and Mason, 2003).**

|  |  | Modelled | |
|---|---|---|---|
|  |  | Water | No Water |
| **Observed** | **Water** | A)  Correct Water (Hits) | B)  Under-prediction (Misses) |
|  | **No Water** | C)  Over-prediction (False Alarms) | D)  Correct No Water (Correct Rejections) |


From this a binary pattern performance measure is used to give a deterministic indication of how
well each LISFLOOD-FP simulated flood map has represented the observed data (Mason, 2003 and
Stephens *et al.*, 2014). We chose to use the Critical Success Index (CSI, equation 1 below) as this
measure does not consider 'correct rejections' (*D* in Table 1) in the calculation (Bates and De Roo,
2000, Horritt et al., 2001a, Aronica et al., 2002) and it weights over- and under-prediction equally (*C*
and *B* respectively).
$$CSI = \frac{A}{A+B+C} \tag{1}$$

If 'correct rejections' were included by the use of a different performance measure the result would
be overly optimistic scores, given the large areas of 'no water' normally observed in a SAR image. All
LISFLOOD-FP simulated flood maps would seem to perform exceptionally well with little to help
differentiate between each simulation.

### 1.4  Parameter identifiability

To determine most likely values for '$r$' and '$n_c$' we follow the technique of Wagener *et al.* (2003) in
applying a dynamic identifiability analysis (DYNIA) to the ensemble of CSI score results. Since the
original DYNIA method was applied to continuous data and not discrete observations some changes
are needed which are described at the end of this section.
The first stage in the DYNIA method is to rescale the 'objective function' (i.e. CSI scores) so that they
add up to one, which is done by dividing each model result by the sum of all scores. Next, computing
the cumulative distribution of the rescaled objective function transforms the objective function into
a support measure which sums to unity - the 'cumulative support' – so that each support measure
may be comparable. To obtain the information content (IC) a confidence limit is applied to the
rescaled objective functions to exclude outliers. The width of the confidence limit depends on how
the best performing parameters are spread within the parameter space: a wide confidence limit
suggests that the parameters are distributed within the parameter space evenly, whereas a narrow
confidence limit suggests that the best performing parameters are located within a smaller range. If
the best performing model parameter combinations are distributed evenly within the parameter





space a confidence limit may be applied to the data. To normalise results, a transformation measure
was used (1 minus the width of the confidence limits over the parameter range, normalised to run
from zero to one): so a value close to 1 is equivalent to a high IC. The IC can have any value between
0 (no information in that observation for parameter identification purposes) and 1 (observation is
most informative for the parameter). The IC results are shown in section 3.2 below.
The second stage in DYNIA is to find the identifiability by locating where in the parameter-time space
most parameter information can be found. This is achieved by examining a plot of 'cumulative
support' against a parameter value. Any deviation from a straight line gradient of this cumulative
support indicates whether the parameter is conditioned by the objective function or not. The
stronger the deviation, the stronger is the conditioning/identifiability of the parameter variable. This
is done using the marginal parameter distributions – interactions are therefore only implicitly
accounted for. The final stage is to organise the data into bins and calculate the gradient of the
cumulative support between them. The results from this examination are shown in section 3.3
below. The IC and identifiability for all single SAR acquisitions are shown along with particular SAR
combinations/groupings: by flood event and by position in the flood hydrograph as detailed in
section 2.2 and Table 3. The identifiability plots have been converted to cumulative distribution
function (cdf) plots for easier cross- comparisons.
The original method proposed by Wagener *et al.* (2003) recommends a pre-selection of models
before stage 1 by using only the top 10% performing models. We deviate from this original method
by using the complete sample of 1000 sets of CSI scores since we found this gave a clearer picture of
identifiability with our data. Also to achieve the 'grouping' of SAR images (section 3.3.2 to 3.3.4) the
CSI scores were multiplied before rescaling combined scores and obtaining the cumulative
distribution at the end of stage one.

## 214   **2   Study area and data used**

The area around Tewkesbury (UK), located at the confluence of the Rivers Severn and Avon is our
test location. Figure 1 illustrates the 30.5 km by 52.4 km model domain, showing the two main rivers
and their tributaries.



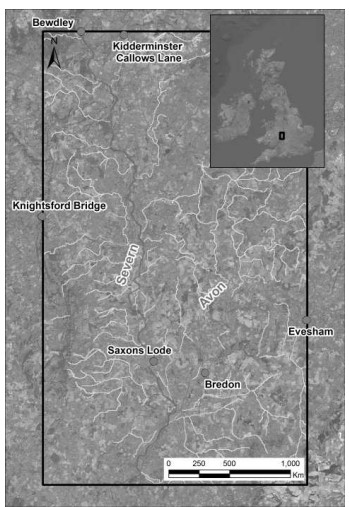


**Figure 1 Extent of the River Severn model**

## 2.1 River Severn model set up

Two separate LISFLOOD-FP models were created to test the methodology. Both models are at 75m
spatial resolution and use the same background DEM. Additionally, both models use the same
gauged inflows and have a rectangular shaped channel. At the lower end of the model a 'free'
downstream boundary condition was applied with a fixed energy slope of 0.00007, based on the
average valley slope.
The differences between the two separate models are in how bankfull channel depth and Manning's
channel roughness values are obtained. First, an 'observed' model was created using surveyed cross
sections of the main rivers to determine channel width and depth with a fixed Manning's channel
roughness parameter of 0.038 (a value representing a main channel which is clear with some
winding and presence of stones/vegetation - from Ven Te Chow, 1959). The cross section survey
data were provided by the Environment Agency of England and Wales (EA). Second, a 'test' model
was created in which the depth parameter '$r$' and Manning's channel roughness parameter '$n_c$' are
determined using the DYNIA identifiability analysis. The depth parameter '$r$' was sampled between
0.0 and 0.5 so that the modelled river depth would never exceed half of the river width. This is a
reasonable assumption for this site where the Severn is on average 60m wide (estimated from LiDAR
data, details below) with surveyed bankfull depth varying between 6m and 11m. The range of
Manning channel roughness values for the sampling was set between 0.015 and 0.100 (Ven Te
Chow, 1959). A low '$n_c$' of 0.015 would represent a channel which is clear and straight whereas and a
high '$n_c$' value of 0.100 would represent a channel with very thick vegetation/submerged branches
present. This range widely encompasses recommended roughness values for the rivers present
within the study domain.
For both the test and observed models the Manning's floodplain roughness value was set at a
standard 0.06 for the entire domain. This is a reasonable average for the floodplain which is mainly
crop and grassland (0.03-0.04) but with presence of some trees (0.12) and brush (0.07). The
Manning's values for the floodplain and the river channel ('$n_c$') are assumed to be spatially and also





temporally invariant. The floodplain topography was taken from a 2m resolution LiDAR based Digital
Surface Model (DSM) with vertical RMSE of 0.10m taken on 9 December 2005 by EA. The EA treated
the DSM to remove structures and vegetation and we then spatially averaged this Digital Terrain
Model (DTM) to 75m resolution as this is an appropriate compromise between model fidelity and
computational cost for rural river reaches (Horritt and Bates, 2001b). The 75m DTM was further
processed to reinsert the maximum height of the flood embankments along the reach in order to
preserve normal flood behaviour along the river banks. No bridges or weirs are included in the
model. Neal *et al.*, 2011 and Garcia-Pintado *et al.*, 2013 provide additional details of the model set
up for the River Severn around Tewkesbury.
Observed flows obtained from the EA were used as inflow to both models. Forcing flows come
principally from the gauging station on the River Severn at Bewdley but with additional inputs from
three tributaries of the River Severn: the Rivers Stour, Salwarpe and Teme. For the River Avon flows
from the Evesham gauging station were used, with two additional flow contributions from the Avon
tributaries Bow Brook and the River Isbourne. A smaller input from a wetland area west of
Tewkesbury was also included, with flows scaled by area from the Salwarpe gauged flows.
The River Severn floods events of March 2007 (simulation period: 19 February 2007 - 29 April 2007),
July 2007 (simulation period: 5 June 2007 - 12 August 2007), January 2008 (simulation period: 26
November 2007 - 25 February 2008) & January 2010 (simulation period: 4 January 2010 – 18
February 2010) were modelled. The dates were chosen so the model would start at least 10 days
before the start of the flood and end after flows had returned to within bank.

### 266 2.2 SAR observations of the River Severn

Historic ENVISAT Wide Swath Mode ('WSM', 150m resolution) data are available from the European
Space Agency's ENVISAT catalogue. Previous research at this site has largely focused on the July
2007 flood event observations (Mason *et al.,* 2012 and 2014, Durand *et al.,* 2014, Garcia-Pintado *et*
*al.,* 2013, Schumann *et al.,* 2011). The present work makes use of many other historic flood
observations in this area – namely the floods of March 2007, January 2008 and January 2010. Details
of the satellite acquisition times are shown in Table 2, along with hydrologic information on the
flood taken from the gauging station at Saxons Lode in the middle of the model domain. Time to
peak describes the number of hours between the start of the event and the peak of the flood.
Flooding from sequential events or with high contributions from other sources such as groundwater
will therefore have a greater time to peak.
**Table 2 The ESA sourced ENVISAT ASAR WSM acquisitions used with equivalent flow and return period data**
**for Rivers Avon and Severn: gauged data was obtained from the EA.**

| SAR ID | Date | Time | Time to flood peak (approx., hrs.) | Gauged flow (m3/s) | Event return period (approx.) | Gauged flow (m3/s) | Event return period (approx.) |
|---|---|---|---|---|---|---|---|
| | | | | At Saxons Lode (Severn) | | At Evesham (Avon) | |
| 1 (March 2007, 1) | 05/03/2007 | 10:27 | 268 | 388 | <5 | 188 | <3 |
| 2 (March 2007, 2) | 05/03/2007 | 21:53 | 268 | 405 | | 87 | |



| SAR ID | Date | Time | Time to flood peak (approx., hrs.) | Gauged flow (m3/s) | Event return period (approx.) | Gauged flow (m3/s) | Event return period (approx.) |
|---|---|---|---|---|---|---|---|
| | | | | At Saxons Lode (Severn) | | At Evesham (Avon) | |
| 3 (March 2007, 3) | 08/03/2007 | 10:34 | 268 | 419 | | 55 | |
| 4 (March 2007, 4) | 08/03/2007 | 21:58 | 268 | 400 | | 45 | |
| 5 (July 2007, 1) | 23/07/2007 | 10:27 | 132 | 532 | 30-40 | 196 | 110-150 |
| 6 (July 2007, 2) | 23/07/2007 | 21:53 | 132 | 512 | | 167 | |
| 7 (January 2008, 1) | 17/01/2008 | 21:55 | 228 | 432 | | 64 | |
| 8 (January 2008, 2) | 24/01/2008 | 10:12 | 228 | 440 | <5 | 28 | <3 |
| 9 (January 2008, 3) | 24/01/2008 | 21:38 | 228 | 433 | | 26 | |
| 10 (January 2010, 1) | 18/01/2010 | 10:30 | 73 | 407 | <3 | 107 | 2 |
| 11 (January 2010, 2) | 18/01/2010 | 21:53 | 73 | 403 | | 37 | |


We separated these 11 SAR observations into different categories by particular flood event (section
3.3.2) or where the acquisition occurs on the flood hydrograph (section 3.3.3). Table 3 records this
segmentation of the 11 acquisitions into categories.

283                                    **Table 3 Description of SAR groupings**

| Description | SAR ID | | | | | | | | | | |
|---|---|---|---|---|---|---|---|---|---|---|---|
| | 1 | 2 | 3 | 4 | 5 | 6 | 7 | 8 | 9 | 10 | 11 |
| **By flood 'event'** | March 2007 | | | | July 2007 | | January 2008 | | | January 2010 | |
| **By point in hydrograph [r =rising limb, p = peak, f = falling limb]** | r | r | f | f | f | f | p | f | f | p | p |


To validate the methodology we take advantage of a very high-resolution (0.2m) aerial photograph
taken by the EA on 24 July 2007, at 11:30 GMT (Giustarini et al., 2013). A flood map shapefile was
created from this imagery by manual definition of the flood boundary. This was then converted and
upscaled to a raster with the same spatial resolution (75m) of theLISFLOOD-FP model results.

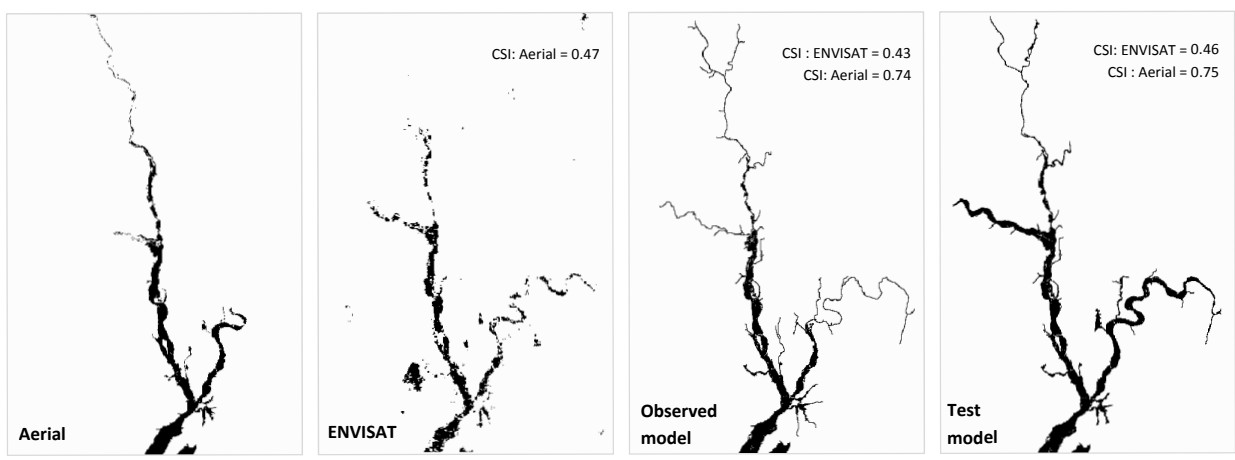

Figure 2 - The July 2007 flood extents as observed by aerial photography (left) and ENVISAT ASAR instruments in WSM (on 23rd at 10:27, centre left). The same flood event simulated in LISFLOOD-FP with surveyed cross sections (centre right, with Manning's channel roughness fixed at 0.038) and the test model with optimally calibrated parameters (right).




## 3 Results and discussion

### 3.1 CSI scores

Figure 2 shows the ENVISAT WSM derived flood map (centre left) from the July 2007 flood event
with the LISFLOOD-FP simulation outcomes from the observed model (centre right) and best
simulated or 'test' model (right) at the same time step. The Critical Success Index (CSI) scores
indicate the ability of the LISFLOOD-FP model to reproduce SAR satellite-derived flood maps like this
one. The CSI score is a scale between 1 (indicating perfect skill in the model) and 0 (indicating no skill
in the model).
The test model CSI results were plotted against the '$r$' and '$n_c$' parameter variables to illustrate CSI
trends with changing parameter value. The figure below compares two plots: one for an ENVISAT
WSM acquisition taken on 23rd July 2007 (10:27am) and one taken on 17th January 2008 (21:55pm).
These CSI plots represent typical results.

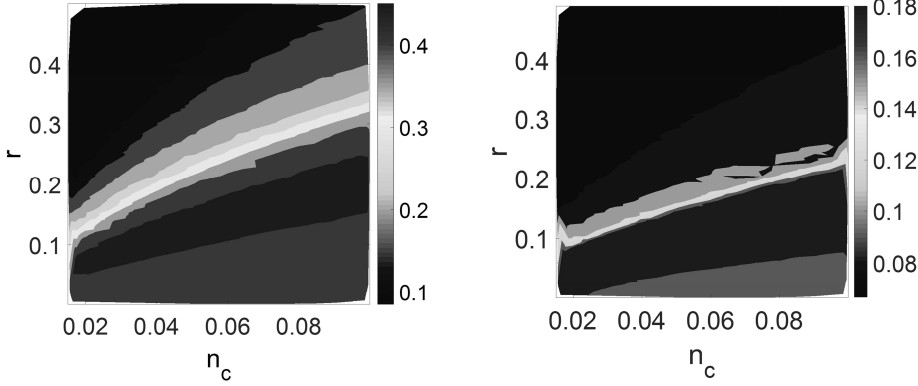

**Figure 3 Single SAR acquisitions are compared with LISFLOOD-FP modelled flood maps for the July 2007 flood event. Left: results from the SAR acquisition on 23rd July 2007 at 10:27, right: result from the SAR acquisition 17[th] January 2008 at 21:55.**

A number of '$r$' and '$n_c$' parameter combinations are able to produce a good result (i.e. equifinality as
described by Beven, 2009). The optimal '$r$' parameter range varies slightly depending on the image
considered. Here test models with the best reproduction of the SAR flood map have '$r$' parameters
between approximately 0.10 and 0.30 (July 2007) and between 0.07 and 0.25 (January 2008).
Generally, the best reproduction of the SAR flood maps is obtained with models that have an '$r$'
value in the lower parameter range and this translates to a wide and shallow river channel and large
width to depth ratio.
For the 23[rd] July 2007 and 17[th] January 2008 events, the top CSI scores for the test model were 0.46
and 0.20 respectively. Whereas the top CSI scores for the observed model were 0.43 and 0.20. To
compare results against those from the same models using the rescaled aerial photograph flood
maps; the best CSI scores were around 0.75 for the test model and 0.74 for the observed model. The
observed and calibrated LISFLOOD-FP models produce CSI scores that are modest by comparison to
the aerial results and other studies, which have used higher resolution SAR imagery for validation





(e.g. Bates *et al.* 2006, Di Baldassarre *et al*. 2009a, Di Baldassarre *et al*. 2010). This is likely
attributable to the incomplete nature of the ENVISAT WSM SAR image and resulting processed flood
map (Figure 2, centre left).  Whereas the aerial photograph (Figure 2, left) has improved
representation of flooding because there are no detection gaps as the flood extent was delineated
manually.
Figure 3 also illustrates the co-variance and a linear dependency between the two parameters.
Although the choice of parameter range emphasizes it, there is a slightly greater skill score
sensitivity to changes in '$r$' than for '$n_c$'. Since changes in channel depth would have an immediate
and local impact on flood level, and hence flood extent, it is logical to see changes in '$r$' producing a
change in flood extent throughout the modelled domain. Channel roughness changes have an
impact more on flow velocities, consequently impacting on the timing of flood wave propagation
through the channel (as discussed in Neal *et al.,* 2015) which would have a more spatially diffuse
impact on flood extent.
Previous SAR based assimilation studies (Hostache *et al.,* 2009, Mason *et al.* 2009, Di Baldassarre *et*
*al.* 2009a) show that with a known and fixed channel bathymetry there is sufficient sensitivity in the
roughness parameter to enable calibration. The above findings indicate that the sensitivity of '$n_c$' is
less obvious when '$r$' is also unknown. There are previous studies also where, as here, channel
friction appears less sensitive when other parameters are simultaneously calibrated. Roux *et al.*
(2008) found sensitivity in hydraulic model response to channel roughness to be weaker than
sensitivity to geometry parameters and boundary conditions within a Generalised Sensitivity
Analysis framework. Additionally Garcia-Pintado *et al.* (2015) found that sensitivity to bathymetry
parameters dominated when using the Ensemble Transform Kalman Filter to simultaneously
estimate bathymetry and channel friction. The sensitivity in channel friction may therefore be not as
obvious when other parameters are simultaneously calibrated because the model is no longer
compensating for previously unrepresented uncertainties. It could be suggested that channel friction
is reverting to its true sensitivity and so when channel friction is combined with more dominant
parameters such as channel bathymetry it is rendered less useful for model calibration.
For '$n_c$', there is not a significant trend and this parameter appears insensitive when estimated
simultaneously with the channel depth: channel roughness can take any value between 0.01 and 0.1
and still yield optimal results as long as '$r$' is also unknown. For this reason the results presented
below now focus only on the more identifiable '$r$' parameter.
## 3.2  Information content (IC)
Table 4 presents IC results for parameter '$r$'. For single SAR observations (left column) there is clearly
greater information content in the July 2007 flood event images. The inundation during this higher
magnitude event extended well into the floodplain and the flood detection algorithm was able to
detect a large number of flooded cells. The lower IC scores for the March 2007, January 2008 and
January 2010 events show that these observations contain less information to help estimate
parameter '$r$'.





**Table 4 - Information content for 'r' from SAR observations and groups of SAR observation with a 90%**
**confidence limit applied.**

| Sequence | Information Content | Sequence | Information Content |
|---|---|---|---|
| 1 - Mar07_1 | 0.10 | Rising limb | 0.13 |
| 2 - Mar07_2 | 0.11 | Peak of hydrograph | 0.23 |
| 3 - Mar07_3 | 0.11 | Falling limb | 0.64 |
| 4 - Mar07_4 | 0.11 | March 07 event | 0.50 |
| 5 - Jul07_1 | 0.16 | July 07 event | 0.37 |
| 6 - Jul07_2 | 0.19 | January 08 event | 0.25 |
| 7 - Jan08_1 | 0.10 | January 10 event | 0.14 |
| 8 - Jan08_2 | 0.11 | All SAR  [1-11] | 0.68 |
| 9 - Jan08_3 | 0.11 | | |
| 10 - Jan10_1 | 0.10 | | |
| 11 - Jan10_2 | 0.10 | | |


Combining images boosts the IC scores considerably as can be seen in the right hand side columns of
Table 4. The image results were combined by multiplying CSI scores for each model for each
combination. Different combinations were tested including grouping according to flood event and
position on the hydrograph as well as 'all SAR' data.
For IC the July 2007 flood now no longer outperforms the rest and instead combinations of images,
like the March 2007 flood event, have greater information on '*r*'.  The March 2007 flood
combination combines observations either side of the hydrograph peak and the January 2008 flood
combination observes flooding 'at peak' and soon after in the falling limb. By contrast the reduced-
scoring January 2010 and July 2007 combinations acquired images at a single stage in the
hydrograph only. We might conclude that the detection quality of the SAR flood maps and timing of
acquisition must influence the final IC score and this is supported also by the observation that the
early 'falling limb' grouping has one of the largest IC scores here.
Nevertheless, the number of SAR flood maps combined appears to be important also since the 'all
SAR' and early 'falling limb' (just over half of these SAR images, Table 3) groupings emerge as
providing the highest IC. The March 2007 flood grouping also contains twice as many members as
the July 2007 or January 2010 flood groupings and outperforms both. Clearly, incorporating data
from multiple observations improves IC since combining SAR images (and CSI scores) improves the
likelihood of extracting information on the unknown parameters. However it is not simply a question
of numbers otherwise 'falling limb' (combining 6 SAR flood maps for an IC score of 0.64) would not
be approaching the success of 'all SAR' (combining 11 SAR flood maps for an IC score of 0.68). Nor is
greater information necessarily revealed by removing poor scorers ('all SAR' IC score reduces from
0.68 to 0.64 when the 4 lowest scoring flood maps are removed from this grouping). Instead the
solution may lie in using SAR flood maps around the peak and falling limb of the flood since
combining 'falling limb' and 'rising limb' observations together yields an IC score of 0.65 but
combining 'falling limb' and 'peak' observations together provides an IC score of 0.67. Further work
and data is necessary to draw any firm conclusions for the '*r*' model parameter.






### 3.3 Identifiability

The identifiability of 'r' within single images and combinations of images is assessed in this section.
This shows where the parameter is most easily identified in the ensemble of model results (for a
particular combination of images). A strong identifiability response would be marked out by having a
narrow shape and peak in the following plots, indicating that the best performing parameters are
concentrated in a small area of the parameter space/group of models. Conversely a wide plot would
indicate weak identifiability and that the best performing models would be those with the 'r'
parameter widely distributed. Figure 3 shows that the best performing model parameter
combinations are distributed fairly evenly within the parameter space so a 90% confidence limit was
applied to the data prior to measuring the gradient of cumulative distribution of rescaled support
values. With these data 20 bins was deemed sufficient to clearly show identifiability.

#### 3.3.1 Individual SAR observations

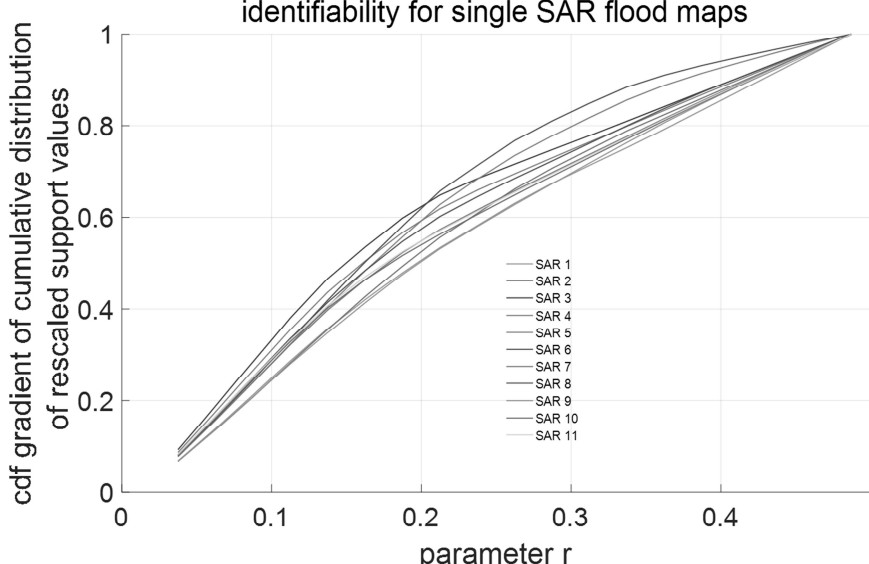

**Figure 4 - Identifiability against 'r' parameter, for each ENVISAT SAR observation in archive.**
Figure 4 shows that typically for the individual SAR observations there is higher identifiability for the
smaller 'r' parameter values. It is evident that the peaks in the majority of these plots occur for 'r'
between 0.05 and 0.15, although there is a subtle difference between the individual results. The first
observations during the March 2007 and January 2008 events are more peaked so that the location
of 'r' can be more easily approximated. By contrast the individual plots for the January 2010 events
are slightly flatter indicating lower identifiability in these SAR observations.

#### 3.3.2 Flood event

This section illustrates identifiability when data from individual SAR images are combined into 'flood
events'. An important characteristic of the 'flood event' identifiability plots is that the SAR
acquisitions are taken together in close sequence. Garcia-Pintado *et al.* (2013) found that a tight
sequence of images could improve model predictions. The identifiability peak for 'flood event'





combined SAR images is slightly narrower such that the '$r$' parameter values/models with higher
identifiability have a range from 0.07 to 0.15. Combining observations in this way appears to focus
the location of the '$r$' parameter more clearly than is possible using single images. However, the
optimum '$r$' value is not stationary and varies between 0.07 to 0.1 and 0.1 to 0.15.

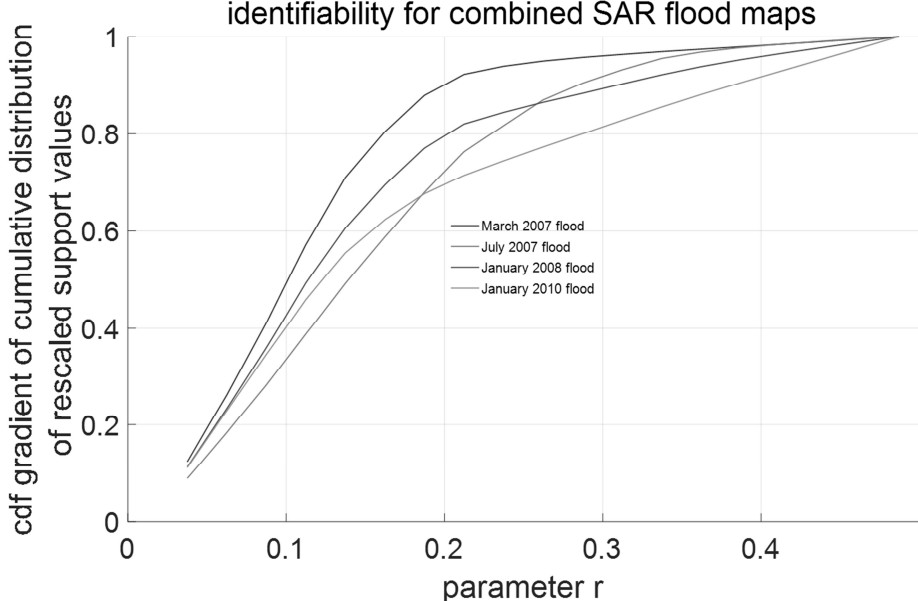


**Figure 5 Identifiability against parameter '$r$', for flood events.**
As also seen in the individual SAR observations, the March 2007 and January 2008 events produce a
strong identifiability. Based just on identifiability, the March 2007 and January 2008 SAR images
might therefore be best utilised to locate the value of parameter '$r$' in the bin 0.10 to 0.125. These
events have approximately the same peak discharge flows at Saxons Lode (see Table 2). However,
the IC results point towards the March 2007 data combination alone as having more parameter
information and the reason for this becomes clear when looking at the binary flood maps contained
in each event. The group of SAR images taken in March 2007 combine to yield a more complete
representation of the flood extent than the combination from January 2008. So although
identifiability shows that both March 2007 and January 2008 flood events would be useful to locate
the parameter '$r$', IC shows the information contained in the March 2007 binary flood maps to be
higher.
### 3.3.3 Through the flood hydrograph
Figure 6 looks at identifiability at three stages of a flood hydrograph for the '$r$' parameter, namely
from observations at the (late) rising limb, the peak and the (early) falling limbs are plotted, with
reference to the stage hydrograph at Saxon's Lode in the central portion of the model domain.
Previous studies have found that the scheduling of SAR images is important for calibration of
models. Di Baldassarre *et al*. (2009b) found that identification of the optimal model parameters
depended on the timing of the SAR image acquisition and the magnitude of the flood event. Garcia-



Pintado *et al.*'s (2013) paper established that to improve forecasting of water levels in a model,
regular observations during the rising limb and then less frequent observations during the falling
limb gave most success. Additionally, Schumann *et al.* (2009b) cautioned that SAR images acquired
during the wetting and drying phases of a flood could be showing floodplain connections and
dewatering processes unconnected with the hydraulics represented by the model.

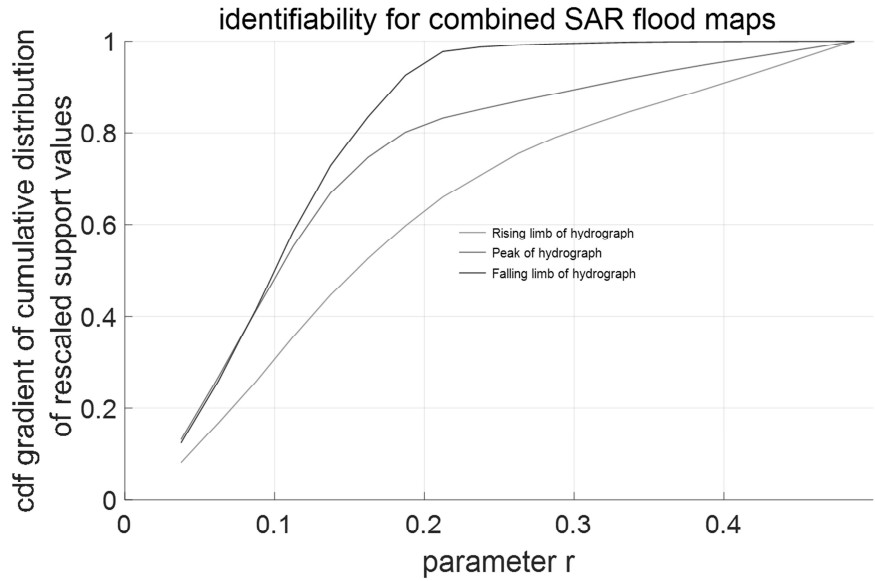


**Figure 6  Identifiability against r parameter, for different stages in hydrograph.**
The number of SAR acquisitions within each category is limited but Figure 6 shows there is a
difference in identifiability for these separate phases and that the location of optimum '*r*' varies. The
narrowest peak and therefore strongest '*r*' parameter identifiability occurs for those images taken
around the flood peak and falling limb of the hydrograph. The weakest identifiability for the '*r*'
parameter occurs for the images taken during the rising limb in contrast to previous studies (e.g.
Garcia-Pintado *et al.,* 2013). The IC results in Table 4 also support this. The reasons for this
disagreement with earlier research may simply lie with the way that 'through the hydrograph'
images were categorised. The method makes use of only a single independent gauge (at Saxons
Lode) to define the phases and as such it could be an oversimplification of the flood dynamics in a
large domain such as this where the 'rising', 'peak' and 'falling limb' of the flood occur at significantly
different times depending on where you measure within the model domain. It might be more
accurate to state that these flood maps around the peak and early falling limb capture the average
moment of transition of flows over banks into the floodplain and these are better conditions for
identifying channel depth parameters.
Alternatively this divergence of findings for the optimum image time could be explained by the
different experimental set up and goals. Garcia-Pintado *et al.* (2013) made use of distributed and
derived water levels to correct model inflow errors and improve model predictions with assimilation,
whereas identifiability here makes use of SAR derived flood extent to calibrate reach-averaged
bathymetry and roughness parameters for the entire river network. Information obtained during the





rising limb was the most useful time to correct inflows because the water level and channel volumes
are most changing during this time. Whereas this experiment, in locating the optimum bathymetry
and roughness parameters, relies on mapping of flood extent (i.e. at bankfull and overbank). This is
seen most usually in the so-named 'peak' and 'falling limb' images where there is indeed flood
extent but also where flows (at some locations within the model domain) are transitioning between
channel and floodplain.

### 3.3.4   All data

Figure 7 shows the identifiability result for all 11 SAR flood maps combined. As for the IC results, this
arrangement produces an observable improvement in identifiability compared with the single SAR or
'flood event' plots. Although a single image does provide the information needed to locate
parameter '$r$', these results show that a group of similarly conditioned images can locate '$r$' more
distinctly and thus with greater confidence. The strongest identifiability is for those models with '$r$'
between 0.10 and 0.12 when looking at 'all data'. These results suggest that greatest information for
parameter '$r$' can be obtained by making use of as much data as is available: in other words that by
simply making use of all available images the depth parameter '$r$' becomes more identifiable.
Moreover 'all data' mixes flood magnitudes and therefore the model is therefore likely to be more
robustly calibrated for a range of event scenarios. In this instance including even relatively poor
flood maps does not negatively impact the result.  However, this might not always be true and
situations may arise where particular flood maps (or sets of flood maps) would be disinformative.

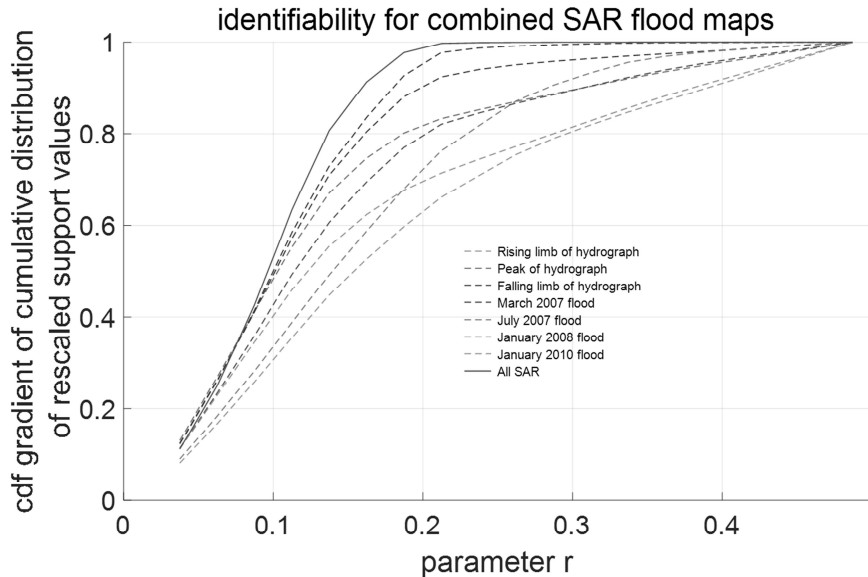


**Figure 7  Identifiability against r parameter, for all hydrographs.**

## 4   Constraining the channel roughness parameter '$n_c$'

The results above show that calibration is possible for the more dominant depth parameter but that
roughness is less easily located in this simultaneous calibration methodology. So far it is assumed
that no ground data are available to give prior information on either parameter and so the ranges




are deliberately broad. However one or both parameters could be constrained further with some
knowledge of the catchment and standard look up tables (e.g. Phillips *et al.,* 2007, Ven Te Chow,
1959). Given that even a cursory examination of Google Earth imagery shows regions of meander
and channel alteration, obstructions and changing vegetation along the River Severn reach, the
Manning's channel roughness values are most likely to lie between 0.035 and 0.055 (rather than in
the wide 0.015 to 0.100 range previously assumed). This section shows that if we constrain the '$n_c$'
parameter to a narrower range based on physical principles and expert judgement it is possible to
improve on first results.

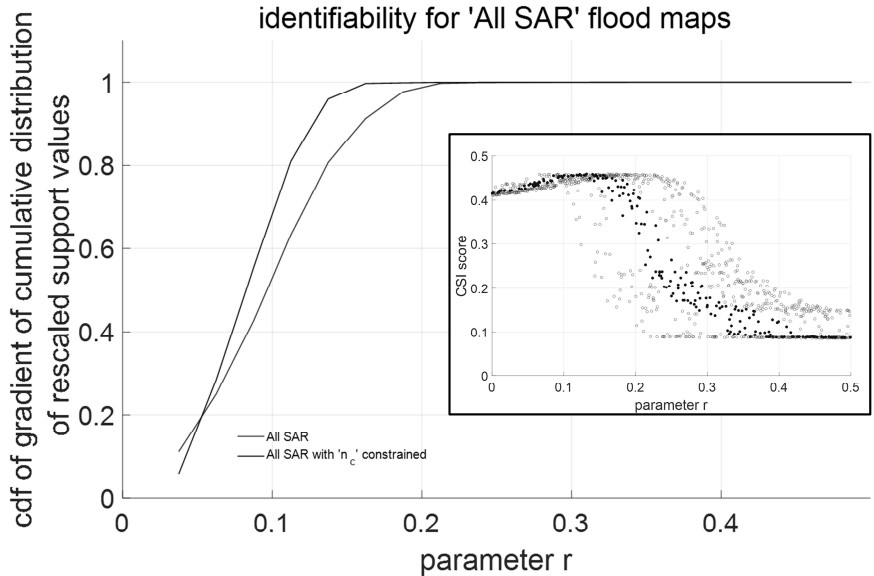

**Figure 8 – identifiability for 23rd July 2007 at 10:27 showing 'all data' (red) and with '$n_c$' restricted to**
**between 0.035 and 0.055 (blue). Inset: the CSI plot against parameter 'r'.**
As before, the calibration location of '*r*' varies marginally with each SAR image or grouping. Figure 8
shows CSI results against parameter '*r*' for a single observation (July 2007) along with the 'all data'
identifiability plot. The 236 models which satisfy the constraint of having '$n_c$' between 0.035 and
0.055 are shown in blue (red shows all other model results).

In this set-up, and focusing just on the top performing models (the maximum CSI score or within 2%
of it), the information rich combinations like 'all data' (Figure 8, right) suggest the location of '*r*' is
between 0.09 and 0.11.  This translates to a reach-average model depth of between 6.4m and 8.5m
and is reasonably close to the observed data. In this group, the single highest scoring model has '*r*' of
0.086 ('$n_c$' of 0.036) and so indicating the optimum reach-average model depth is around 6.51m. The
equivalent rectangular depth from the EA survey is 5.63m (assuming a reach median width of 76m)
using bank-full cross sectional area. The difference therefore between the calibrated value and the
observed equivalent is approximately 0.88m (an error of 16%).

With '$n_c$' constrained the '*r*' value moves to a lower depth range. The model responds to this specific
channel friction by altering the speed of the flood wave and flow velocities. With '$n_c$' constrained to





0.036, Saxons Lode station experiences the 'flood peak' closer to the observed peak time than
models with other friction values. These results highlight the important reasons for calibrating this
second parameter concurrently. If channel roughness were set too high the flood wave would be
delayed. Set too low and the flood wave would be too advanced.

## 5    Conclusion

This paper presents a methodology for dual calibration of bankfull depth and channel roughness
parameters of the LISFLOOD-FP Sub-Grid hydraulic model using SAR data and a binary pattern
classification measure based on flood extent. Multiple models performed well initially, but by
employing an identifiability methodology we located the area of the parameter space with highest
information for the depth parameter '$r$'. The location narrows with the use of more SAR images.
The methodology provides some information on which single and combinations of SAR flood maps
would be most useful for calibration purposes. Single SAR flood maps would be sufficient to calibrate
the depth parameter but the identifiability is much improved when multiple maps are combined.
Combinations aligned according to particular flood events/magnitudes are not conclusively different,
but using many or all available SAR images does offer a real improvement in identifiability. There are
indications that combining maps with similar flood duration, or stage of flood (i.e. SAR images
acquired close to peak or just after) would be beneficial for calibrating the reach-average depth
parameter, but further work is needed with more targeted observations than the 11 used here. For
robustness, a good range of flood magnitudes should be used for calibration.
The channel roughness parameter '$n_c$' was less sensitive to variations in flood extent and we failed to
locate a representative value for this parameter when '$r$' was also varied. The likely cause probably
due to the initial range selected being too broad and the suggestion that depth/bathymetry is the
more dominant parameter in the model which largely overrides, at this model scale at least, the
significance of channel friction. By constraining '$n_c$' to a more plausible range it was possible to
improve the calibration method and further improve the global estimate for the depth parameter.
Under this constraint the models with top CSI and identifiability results show that the reach-
averaged depth parameter is calibrated to 0.086, translating roughly to a reach-average depth of
approximately 6.51m. This is an error of 0.88m compared with an equivalent measure from
observed cross section data, where channel depth is approximated as 5.63m.
A benefit of this methodology is that although we used gauged inflows within the model, in theory
the calibration methodology should work also with no recourse to ground data if good inflows can
be simulated and a good DEM is available. The method also does not require a step to obtain water
levels from the flood data. It does however make some simplifications and assumptions. First, the
method assumes that there are no errors in the return signals or processing of the ENVISAT WSM
images the derived flood maps therefore represent the true and full flood extent, however in reality
all data have error. There is also error likely in the assumptions behind the model set up. Neither has
the importance of the SAR resolution been tested here. Second, we assume that the friction and
depth parameters are uniform through the model domain when in reality spatial variability will exist.
The calibrated parameters here are therefore reach-averaged values and it is for this reason that the
methodology is perhaps more appropriately used for medium sized catchments with ostensibly
negligible variation in domain channel width. For width varying and large catchments, future work



will investigate the impact of applying the methodology within smaller sub-reaches (i.e. 'sub-regions'
or tributaries) where hydraulics and hydrology are similar.

## 6  Acknowledgements

We thank the Environment Agency of England and Wales for providing the river cross-section data,
DEM and gauging station data.
M. Wood's contribution was supported by the National Research Fund of Luxembourg through the
PAPARAZZI project (CORE C11/SR/1277979).



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
