# Peer review of "Calibration of channel depth and friction parameters in the LISFLOOD-FP hydraulic model using medium resolution SAR data"

_Hydrology and Earth System Sciences, 2015_

## Referee Comment (RC1) · Anonymous Referee #1 · 15 Feb 2016

Review of the paper:

Calibration of channel depth and friction parameters in the LISFLOOD-FP hydraulic model using medium resolution SAR data By M. Wood, R. Hostache, J. Neal, T. Wagener, L. Giustarini, M. Chini, G. Corato, P. Matgen & P. Bates

The paper presents a method to calibrate simultaneously the bankfull channel depth and channel roughness parameters within a 2D LISFLOOD-FP hydraulic model using an archive of moderate (75m) resolution SAR satellite-derived flood extent maps and a binary performance measure for a 30x50km domain covering the confluence of the rivers Severn and Avon in the UK. The unknown channel parameters are located by a novel technique utilizing the Information Content and identifiability, (previously devel-

oped by one of the authors of the manuscript) of the single and combinations of SAR flood extent maps to find the optimum images for model calibration.

GENERAL COMMENTS

Overall, I found the paper very interesting and adequate for the publication in the HESS journal. The importance of the understanding of the value of single or multiple SAR images in the identification of hydraulic model parameters is very important since – due the ever greater loss of ground information – the use of satellite images will be increasing in the near future. However, I found many drawbacks mainly due to the paper organization and presentation that have to be solved by the authors before it can be considered adequate for a publication in the HESS journal. Moreover there are some key points that I would like to underline.

One is the justification of the small sensitivity of the roughness parameter with respect to the channel bathymetry that to me seems reasonable but I found some difficulty in understanding if this is supported by a robust analysis or it is not adequately shown in the manuscript. To this end, it would be interesting to see the DNYA analysis carried out also for the channel roughness in order to recognize its information content and its value of identifiability. If this was already done, but not shown, some comments or an explaining figure would be very welcome.

A second point is the assumption that the error related to the processing of the SAR image will not affect the results (not considered for simplicity). I thinks that these errors are part of the procedure of identifiability and are able to affect the information content of the different images. For this point my question is: due to the different acquisition, times of the SAR images under different atmospheric and land conditions can be considered the error related to the image processing stationary? My opinion is that this error varies from image to image. This at least deserves some discussion. The authors could consider that aerial flood maps for analyzing this point or, since the area is very well instrumented, doing the same type of analysis with stage data.

[Figure]

A final point is the quality of all the figures in the manuscript that I found very poor and such that to impede a proper understanding of the manuscript.

Based on that I recommend publication after major revisions. In the following, the authors can find a list of comments with the associated relevance listed in order of appearance in the manuscript.

COMMENTS

Pag. 2 Lines 53-76 MINOR: the authors may also cite the work of Moramarco et al. (2013) which uses an interesting method for identifying the flow depth distribution in natural channels.

Moramarco, T., Corato, G., Melone, F., Singh, V.P., 2013. An entropy-based method for determining the flow depth distribution in natural channels. Journal of Hydrology, 497,176-188.

Pag. 3 Lines 88-91 MINOR: Can you rephrase this sentence more clearly?

Pag. 4 Lines 120-128 MAJOR: If I understand correctly the channel depth is expressed as H=r*B where H is the channel depth, and B is the width of the channel. Since the hypothesis of linear scaling is central in the study, I thinks this part deserves more profound discussion about: 1) How much it will affect the results of the study. 2) Which are the expected problematics associated with the uniform channel depth.

See also the paper of Yan et al. (2014) where H is a free parameter of the model uniform along the river reach.

Yan, Kun, et al. "Exploring the potential of SRTM topography and radar altimetry to support flood propagation modeling: Danube case study." Journal of Hydrologic Engineering 20.2 (2014): 04014048.

Pag 3 Section 1.2. MINOR: A scheme or figure of the method would significantly help to understand the image-processing algorithm.
[Figure]

Pag. 6 Lines 211-213: MODERATE: it is not clear how the procedure is used with multiple images. Please provide more details.

Pag 7 Figure 1 MINOR: The quality of this figure is very poor. Please provide a larger and cleared picture where the identification of the study area and the boundary conditions are more clearly visibile.

Pag 11 Figure 2 MODERATE: the quality and the description of this figure is very poor. Also, ENVISAT and Aerial data seem to be a bit different although with this picture is very difficult to compare the results. I understand that the processing of the SAR image inherently contain errors, I am wondering if the results of the paper might be affected by these errors. The authors could test the procedure also on the aerial photograph to understand the effect of the errors in the processing of the image or on the observed stages.

The observed model which is expected to behave better than the test model seems to be worse than the test model? Do you have a justification for that? Does this depend on the calibration?

Pag 11 Figure 3 MAJOR: Please provide a better figure with colors. It is very difficult (with this figure) to follow the authors' statements.

Pag 11 Section 3.1 MODERATE: I found this section very difficult to follow and to read. I suggest to try to present it better.

Pag 12 lines 346-349, MAJOR: the authors concluded that nc is insensitive when estimated simultaneously with the channel depth. However,it seems that this was concluded based only on two images (23rd July 10:27, and 17 January 2008 21:55). Do the authors exclude that this is true in any case and there are not effects of the time of acquisition and the magnitude of the flood event? I think the authors should provide more proofs for this statement. Overall, I find this assumption reasonable however I think that including the DYNIA also for the parameter nc would add a lot of value to the

paper.

Pag. 13 line 361. MODERATE: It is not clear how the IC score is calculated for multiple images.

Figures 4,5, 6, 7 and 8 MAJOR: The interpretation of these figures must be described in the method section. From the text it is very difficult to follow the authors' statements. Please provide a better quality figures as well. It is impossible to discriminate between the different lines. If I understand correctly these are the cumulative distribution of the rescaled support values and not the gradient. The gradient should refer to their slopes. Isn't? If so, I expect a figure like the one in the paper of Wagener et al. (2003), (see FIGURE 8 in their paper).

Pag 18 lines 503. MODERATE: No colors can be seen in the figures.

Pag 19 lines 545-556. MODERATE: I expect here some discussions about the possible consequences of the assumptions made in the paper.
* * *

---

## Referee Comment (RC2) · G. J.-P. Schumann (Referee) · 21 Feb 2016

G. J.-P. Schumann (Referee)

gjpschumann@gmail.com

This paper describes the use of multiple low resolution SAR images to calibrate channel depth and roughness of a 2D LISFLOOD-FP model of the Severn river floods. The method is based on the identifiability approach using a confusion table performance metric.

The study demonstrates that multiple SAR images albeit low resolution can lead to greater identifiability in the presence of more than one uncertain model parameter. It also cancludes that depth parameter is more sensitive than roughness, which is welcomed since bathymetry can be easier validated than roughness.

The paper is well written and athough it is quite lengthy it is an interesting read and should be publishable after some moderate revisions, particularly related to my last point below. Note that although I selected major revisions, I think the gravity of my concerns would point rather to moderate. I believe they can be addressed without major difficulties.

- The IC approach is really nice and gives an objective assessment of the value of a flood image for calibration. However, what we are still missing in the literature is to find a way that gives an objective IC of a SAR flood map without the need to calibrate first. In other words, in this paper, which I think has a lot of merit, IC is built up based on parameter identifiability rather than for instance inter-comparing each SAR image and applying the score and identifiability that way, so without the need of a model and its parameter but I understand that this is outside the scope of this paper.

- I also think that what is innovative here is the analysis of IC and identifiability in relation to what stage in the hydrograph we are looking at and what type of data we use (single image, combined images, gauge data). I wonder if the title and the introduction should better reflect that since to me this is one of the first papers to try and answer these questions using real data.

- My biggest reservation in this study lies with the choice of performance metric used, which may explain in my opinion why the greatest information content is in the SAR images closest to peak flow. Stephens et al. (2014) showed that the performance measure used here is particularly biased towards largest flooded area (in other words, it always gives the highest score to the biggest flooded area). This is significant in this study and could lead to an unwanted "bias" in the calibration. I suggest the authors repeat the exercise offline with the "F2" measure for instance $((A-B)/(A+B+C))$ or an area in error index $((B+C)/(A+B+C+D))$ to see if the same SAR images give the highest sensitivity still.

---

## Author Comment (AC1) · 26 Sep 2016

**General comment 1:**

I found many drawbacks mainly due to the paper organization and presentation that have to be solved by the authors before it can be considered adequate for a publication in the HESS journal. Moreover there are some key points that I would like to underline.

One is the justification of the small sensitivity of the roughness parameter with respect to the channel bathymetry that to me seems reasonable but I found some difficulty in understanding if this is supported by a robust analysis or it is not adequately shown in the manuscript. To this end, it would be interesting to see the DYNIA analysis carried Printer-friendly version

out also for the channel roughness in order to recognize its information content and its value of identifiability. If this was already done, but not shown, some comments or an explaining figure would be very welcome.

**Response:**

We did indeed carry out the DYNIA analysis on the channel roughness 'nc' parameter for all the SAR images used in this study but we found the results for this parameter were not as sensitive as those for the depth parameter 'r'. Variations in channel roughness value did not produce a strong enough response in the model to match observed flood extent. For this reason these 'r' results did not feature in the final version of the paper. The authors have amended the text in section 3.1 to explain this:

"Consequently an important result of this paper is that - in this particular experimental set up with channel roughness parameter 'nc' examined simultaneously with the channel depth parameter 'r' for the available ENVISAT SAR data - 'nc' has a much reduced sensitivity compared with the 'r' depth parameter response. It is observed that 'nc' will yield optimal results for as long as 'r' is also unknown. This lack of sensitivity of channel roughness in this and all subsequent results meant that 'nc' could not be identified with any real confidence with this methodology (while 'r' is also unknown). So while 'nc' analysis was carried out we present here onwards only those results from the more identifiable 'r' parameter. 'nc' results are now omitted (but can be provided upon request if of interest)".

To illustrate we show some of the channel roughness results in Figures 1 and 2. Here we show firstly the results for the 'r' parameter (red), then 'nc' (blue) CSI results, against parameter value for a sample of the SAR images. These CSI plots generally reflect the results we observed in all SAR data and illustrate the greater sensitivity of the depth parameter over the channel roughness parameter in our experimental results.

Additionally, Figure 3 shows the cdf plot of the gradient of the cumulative distribution of rescaled support values for groupings of SAR data, against 'nc' value. This illustrates

HESSD
how different/lower the results are for 'nc' identifiability when compared with 'r' results (paper fig 7, inset).

Lastly, the Information Content (IC) results for 'groupings' of SAR images are also shown here for 'nc' and 'r' in Table 1. These illustrate how much lower the 'nc' information content can be compared with the IC of the depth parameter, for the same combination of data.

**General comment 2:**

A second point is the assumption that the error related to the processing of the SAR image will not affect the results (not considered for simplicity). I think that these errors are part of the procedure of identifiability and are able to affect the information content of the different images. For this point my question is: due to the different acquisition, times of the SAR images under different atmospheric and land conditions can be considered the error related to the image processing stationary? My opinion is that this error varies from image to image. This at least deserves some discussion. The authors could consider that aerial flood maps for analyzing this point or, since the area is very well instrumented, doing the same type of analysis with stage data.

**Response:**

This is a good point and certainly each SAR image will have a unique processing error associated with it and indeed the errors inherent in the processed SAR data will be passed on to the final identifiability and IC score for single images. These will not be stationary errors, they will vary between images. For example, while different atmospheric conditions will not significantly affect the radar signal, different incident angles can have an effect. In the paper these were considered to be so small compared with the errors associated with the assumptions around parameter identifiability that they were thought to be overshadowed. Furthermore the use of moderate resolution flood

HESSD
imagery for hydraulic model calibration may lead to inaccuracies but it was deliberately chosen because we want to understand the usefulness of this data for global locations where other data may quite simply not be available. And while the magnitude of flooding and the land surfaces affected can cause specific errors/uncertainties, this is somewhat mitigated by the larger spatial scales employed in this analysis.

Other errors in the processing of the SAR image can be evident, such as from bias where in some areas radar data does not inform on the extent of flooding. Ideally, such non-informative areas would be masked out but this requires more comprehensive analysis and is currently an active area of research (e.g. Giustarini et al., submitted: puts forward the idea of flood 'probability' maps to illustrate confidence in the detected flood extent). There is now more discussion about this around Figure 3 in the paper which illustrates the CSI scores for the aerial flood map image that was acquired in July 2007. However since the authors already have some concerns about the length of the document, and the comments of reviewer #2 on the length of the paper also, we attempt to keep these discussions brief.

**General Comment 3:**

A final point is the quality of all the figures in the manuscript that I found very poor and such that to impede a proper understanding of the manuscript.

**Response:**

Thank you for this feedback, we have replaced the figures mentioned with the full colour versions and increased the image resolution to improve the quality.
Based on that I recommend publication after major revisions. In the following, the authors can find a list of comments with the associated relevance listed in order of appearance in the manuscript.

**## COMMENTS:**

**1:**

Pag. 2 Lines 53-76 MINOR: the authors may also cite the work of Moramarco et al. (2013) which uses an interesting method for identifying the flow depth distribution in natural channels.

Moramarco, T., Corato, G., Melone, F., Singh, V.P., 2013. An entropy-based method for determining the flow depth distribution in natural channels. Journal of Hydrology, 497,176-188.

**Response:**

This reference has been added to the paper.

**2:**

Pag. 3 Lines 88-91 MINOR: Can you rephrase this sentence more clearly?

**Response:**

Lines 88-91 have been rephrased to: "In particular the methodology uses flood extent with an accuracy-scoring method that disregards the correct detection of 'no water' pixels"

Pag. 4 Lines 120-128 MAJOR: If I understand correctly the channel depth is expressed as H=r\*B where H is the channel depth, and B is the width of the channel. Since the hypothesis of linear scaling is central in the study, I thinks this part deserves more profound discussion about: 1) How much it will affect the results of the study. 2) Which are the expected problematics associated with the uniform channel depth.

HESSD
**3:**

**Response:**

Extra discussion has been inserted at the end of the manuscript around the limitations of our assumptions and what affect these might have on the results.

**# 4:**

See also the paper of Yan et al. (2014) where H is a free parameter of the model uniform along the river reach.

Yan, Kun, et al. "Exploring the potential of SRTM topography and radar altimetry to support flood propagation modeling: Danube case study." Journal of Hydrologic Engineering 20.2 (2014): 04014048.

**Response:**

This reference has been added to the paper.

**# 5:**

Pag 3 Section 1.2. MINOR: A scheme or figure of the method would significantly help to understand the image-processing algorithm.

**Response:**

A good point, the authors have inserted a new figure to illustrate the steps of the methodology: Figure 1.

**6:**
Pag. 6 Lines 211-213: MODERATE: it is not clear how the procedure is used with multiple images. Please provide more details.

**# Response:**

A description of the procedure for combining image results has been updated in section 1.4 of the paper: "These group scores are determined by multiplying each single model/SAR flood map CSI result with the CSI score of the next SAR flood map until all members of the particular group have been added. The unique combinations which comprise these groups are described in Table 3 below. This combining of CSI scores is done for results from each of the 1000 models/parameter scenarios. The next step is the same as for single CSI scores as described above – i.e. to rescale the objective function and compute the cumulative support".

**# 7:**

Pag 7 Figure 1 MINOR: The quality of this figure is very poor. Please provide a larger and cleared picture where the identification of the study area and the boundary conditions are more clearly visible.

**Response:**

Thank you for pointing this out, Figure 1 (now Figure 2) has now been amended to be clearer with boundary locations highlighted.

**# 8:**

Pag 11 Figure 2 MODERATE: the quality and the description of this figure is very poor. Also, ENVISAT and Aerial data seem to be a bit different although with this picture is very difficult to compare the results. I understand that the processing of the SAR image inherently contain errors, I am wondering if the results of the paper might be affected by these errors. The authors could test the procedure also on the aerial photograph to understand the effect of the errors in the processing of the image or on the observed HESSD
stages.

The observed model which is expected to behave better than the test model seems to be worse than the test model? Do you have a justification for that? Does this depend on the calibration?

**Response:**

A well observed point for Figure 2 (now Figure 3) in the paper. The description of this figure in the text has been updated so it is clearer why it was inserted. Also the CSI scores embedded in the figure have been moved to a table for easier comparison/interpretation for the reader. The figures themselves (particularly the modelled and ENVISAT flood extents) have a coarse resolution that does not reproduce very nicely on the page unfortunately.

This particular aerial flood extent image was derived from a single aerial photograph of the flood of July 2007 on the Severn by manual delineation and so is restricted both by the limits of the photographs (cutting off the upper River Avon for example) and interpretation of the image in terms of the flood boundary through vegetation. In this case we used the aerial data here more for validation purposes rather than explicitly to test the calibration methodology. The aim of the paper was to test a series of more 'moderate' resolution imagery that is more extensively and frequently available.

However the authors have added some additional text around the new table of aerial versus ENVISAT CSI scores to explain further the test results we found when applying the methodology on the single aerial flood extent data. A good comparison could be made between the methodology applied to the ENVISAT observation (acquired 23rd July 2007, 10:27am) and the subsequent aerial image (acquired 24th July 2007, 11:30am) as they were observing the flood merely 24hrs apart. While it is quite likely that the impact of SAR processing errors would be manifest in the final results, and be most obvious when compared with better resolution data such as can be obtained from a gauge record or aerial imagery, the authors felt that an explanation of the impact of
these errors was beyond the scope of the paper (which is rather long already).

In Figure 2 (now Figure 3), it is obvious that the observed model (constructed using surveyed cross sections) has not represented observed flood extent as well as the test model. This is most evident in the tributaries to the main River Severn. We updated the text in section 3.1 of the paper to explain this: "The scores and flood extent for the observed model are not better than the test model results as might be expected. This may be explained by the fact that while the bathymetry of the observed model does come from survey data, the (domain-average) channel roughness value is not calibrated in either model. While the test model had 1000 parameter-varying depth and roughness values, the observed model had a best estimate of domain-average channel roughness parameter (of 0.038). While appropriate for the main rivers, it is evident that the channel roughness value is not suitable for the narrower tributaries."

**# 9:**

Pag 11 Figure 3 MAJOR: Please provide a better figure with colors. It is very difficult (with this figure) to follow the authors' statements.

**Response:**

This contour plot has now been updated in black and white so it is easier to read with two single opposing colours than the original colour spectrum which made it difficult to see where CSI was highest and lowest, even in full colour (see fig: contour plots)

**10:**

**Response:**
Pag 11 Section 3.1 MODERATE: I found this section very difficult to follow and to read. I suggest to try to present it better.

A re-write of this section has been carried out and the authors hope it is now easier to read and follow.

**# 11:**

Pag 12 lines 346-349, MAJOR: the authors concluded that nc is insensitive when estimated simultaneously with the channel depth. However, it seems that this was concluded based only on two images (23rd July 10:27, and 17 January 2008 21:55). Do the authors exclude that this is true in any case and there are not effects of the time of acquisition and the magnitude of the flood event? I think the authors should provide more proofs for this statement. Overall, I find this assumption reasonable however I think that including the DYNIA also for the parameter nc would add a lot of value to the paper.

**# Response:**

This paragraph has been rewritten within the context of the above comment on section 3.1 and hopefully now better explains why 'nc' results were excluded from the final version of the paper. All available SAR data were analysed for 'r' and 'nc' though only two 'r' plots featured in the paper for simplicity. As the authors hope has been explained sufficiently within the general comments section above, the 'nc' parameter did indeed undergo the same DYNIA analysis as the depth parameter but results were not exceptional enough to be included within an already long paper. The same lack of responsiveness of this parameter in the results were observed for all SAR data analysed, and even when the SAR data were 'grouped' the identifiability and IC results did not greatly improve for 'nc'. With this available dataset of ENVISAT SAR images and hydraulic model set up, the authors have concluded that 'nc' is insensitive when estimated simultaneously with the channel depth parameter 'r'. However additional testing would need to be carried out to conclusively say that this insensitivity is true with other SAR data and magnitudes of flooding.
**# 12:**

Pag. 13 line 361. MODERATE: It is not clear how the IC score is calculated for multiple images.

**# Response:**

The grouping of SAR data occurs after each single model/SAR flood map is assessed and given a CSI score. It is the CSI scores which are multiplied together in particular combinations (as described in Table 3) before the DYNIA methodology is applied and identifiability and IC determined. A group IC score is derived from the grouped SAR data and CSI scores multiplied together. This explanation has been inserted in to the paper in section 3.2 and 1.4 for greater clarity.

**# 13:**

Figures 4, 5, 6, 7 and 8 MAJOR: The interpretation of these figures must be described in the method section. From the text it is very difficult to follow the authors' statements. Please provide a better quality figures as well. It is impossible to discriminate between the different lines. If I understand correctly these are the cumulative distribution of the rescaled support values and not the gradient. The gradient should refer to their slopes. Isn't? If so, I expect a figure like the one in the paper of Wagener et al. (2003), (see FIGURE 8 in their paper).

**Response:**

Thank you for pointing out how difficult these figures are to read. The original figure coloured lines have been reinstated in this revised version and an explanation of how the cdf plots should be interpreted is now inserted into the method section.

It is correct that Figures 4 - 8 in the paper are representative of the (gradient of the)
cumulative distribution of rescaled support values. At the suggestion of Prof. Wagener the gradient plots were converted to cumulative distribution function (cdf) plots, and this is what was shown in the final version of the paper. This was a change introduced to make the plots easier to read together and normalised, when the original histogram plots were numerous and previously side-by-side for each grouping. For this experiment, the SAR data acquisitions are not regular or plentiful through time (as in the experiment of Wagener et al., 2003 which used 6 years of continuous daily flow data) and so it was more difficult to represent in a line plot. However for purposes of explanation, the last cdf plot in the paper is converted into a 'gradient' plot for parameter 'r' below (top figure) to mimic Figure 8 in the paper of Wagener et al., 2003. In the opinion of the authors this reconfiguration to 'gradient' appears less smooth but nevertheless still conveys the same information as the original figure (bottom) of 'cdf of gradient' that features in the paper. For this reason, we have not amended the Figures 4-8:

**14:**

Pag 18 lines 503. MODERATE: No colors can be seen in the figures.

**Response:**

Figure colours have been reinserted.

**# 15:**

Pag 19 lines 545-556. MODERATE: I expect here some discussions about the possible consequences of the assumptions made in the paper.

**Response:**

This consequences of our assumptions in the paper have been expanded within the results and discussion sections, as described above.
Interactive

comment

Fig. 1. Figure 1 - Plot of 'r' parameter against Critical Success Index score for left: 23rd July

2007 at 10:27 and right: 24th January 2008 at 10:12.
0.2

0.15 S S S 0.1

0.05

0.05

n parameter

0.1

0.1

0.4 8.00 S S S S S S S S

0.1

0L 0

0.05

n parameter
**Fig. 3.** Figure 3 - cdf of the gradient of the cumulative distribution of rescaled support values for each individual SAR image, against 'nc' parameter value.

---

## Author Response (AR1)

**Anonymous Referee #1 -**

**GENERAL COMMENTS**

*I found many drawbacks mainly due to the paper organization and presentation that have to be solved by the authors before it can be considered adequate for a publication in the HESS journal. Moreover there are some key points that I would like to underline.*

*One is the justification of the small sensitivity of the roughness parameter with respect to the channel bathymetry that to me seems reasonable but I found some difficulty in understanding if this is supported by a robust analysis or it is not adequately shown in the manuscript. To this end, it would be interesting to see the DYNIA analysis carried out also for the channel roughness in order to recognize its information content and its value of identifiability. If this was already done, but not shown, some comments or an explaining figure would be very welcome.*

We did indeed carry out the DYNIA analysis on the channel roughness '$n_c$' parameter for all the SAR images used in this study but we found the results for this parameter were not as sensitive as those for the depth parameter '$r$'. Variations in channel roughness value did not produce a strong enough response in the model to match observed flood extent. For this reason these '$r$' results did not feature in the final version of the paper. The authors have amended the text in section 3.1 to explain this:

> "Consequently an important result of this paper is that - in this particular experimental set up with channel roughness parameter '$n_c$' examined simultaneously with the channel depth parameter '$r$' for the available ENVISAT SAR data - '$n_c$' has a much reduced sensitivity compared with the '$r$' depth parameter response. It is observed that '$n_c$' will yield optimal results for as long as '$r$' is also unknown. This lack of sensitivity of channel roughness in this and all subsequent results meant that '$n_c$' could not be identified with any real confidence with this methodology (while '$r$' is also unknown). So while '$n_c$' analysis was carried out we present here onwards only those results from the more identifiable '$r$' parameter. '$n_c$' results are now omitted (but can be provided upon request if of interest)".

To illustrate we show some of the channel roughness results below. Here we show firstly the results for the '$r$' parameter (red), then '$n_c$' (blue) CSI results, against parameter value for a sample of the SAR images. These CSI plots generally reflect the results we observed in all SAR data and illustrate the greater sensitivity of the depth parameter over the channel roughness parameter in our experimental results:

[Figure]

**Figure 1 Plot of 'r' parameter against Critical Success Index score for left: 23rd July 2007 at 10:27 and right: 24th January 2008 at 10:12.**

[Figure]

**Figure 2 Plot of '$n_c$' parameter against Critical Success Index score for left: 23rd July 2007 at 10:27 and right: 24th January 2008 at 10:12.**

Additionally, Figure 3 below shows the cdf plot of the gradient of the cumulative distribution of rescaled support values for groupings of SAR data, against '$n_c$' value. This illustrates how different/lower the results are for '$n_c$' identifiability when compared with '$r$' results (paper fig 7, inset).

[Figure]

**Figure 3 cdf of the gradient of the cumulative distribution of rescaled support values for each individual SAR image, against '$n_c$' parameter value.**

Lastly, the Information Content (IC) results for 'groupings' of SAR images are also shown here for '$n_c$' and '$r$' in Table 1. These illustrate how much lower the '$n_c$' information content can be compared with the IC of the depth parameter, for the same combination of data.

**Table 1 Information Content (IC) for groupings of SAR images, showing results for $r$ and $n_c$.**

| 'grouping' | Parameter '$r$' | Parameter '$n_c$' |
|---|---|---|
| Rising limb | 0.13 | 0.11 |
| Peak of hydrograph | 0.23 | 0.12 |
| Falling limb | 0.64 | 0.16 |
| March 07 event | 0.50 | 0.14 |
| July 07 event | 0.37 | 0.14 |
| January 08 event | 0.25 | 0.12 |
| January 10 event | 0.13 | 0.11 |
| All SAR   [1-11] | 0.68 | 0.16 |

*A second point is the assumption that the error related to the processing of the SAR image will not affect the results (not considered for simplicity). I think that these errors are part of the procedure of identifiability and are able to affect the information content of the different images. For this point my question is: due to the different acquisition, times of the SAR images under different atmospheric and land conditions can be considered the error related to the image processing stationary? My opinion is that this error varies from image to image. This at least deserves some discussion. The authors could consider that aerial flood maps for analyzing this point or, since the area is very well instrumented, doing the same type of analysis with stage data.*

This is a good point and certainly each SAR image will have a unique processing error associated with it and indeed the errors inherent in the processed SAR data will be passed on to the final identifiability and IC score for single images. These will not be stationary errors, they will vary between images.

For example, while different atmospheric conditions will not significantly affect the radar signal, different incident angles can have an effect. In the paper these were considered to be so small compared with the errors associated with the assumptions around parameter identifiability that they were thought to be overshadowed. Furthermore the use of moderate resolution flood imagery for hydraulic model calibration may lead to inaccuracies but it was deliberately chosen because we want to understand the usefulness of this data for global locations where other data may quite simply not be available. And while the magnitude of flooding and the land surfaces affected can cause specific errors/uncertainties, this is somewhat mitigated by the larger spatial scales employed in this analysis.

Other errors in the processing of the SAR image can be evident, such as from bias - where in some areas radar data does not inform on the extent of flooding. Ideally, such non-informative areas would be masked out but this requires more comprehensive analysis and is currently an active area of research (e.g. Giustarini *et al.,* submitted: puts forward the idea of flood 'probability' maps to illustrate confidence in the detected flood extent). There is now more discussion about this around Figure 3 in the paper which illustrates the CSI scores for the aerial flood map image that was acquired in July 2007. However since the authors already have some concerns about the length of the document, and the comments of reviewer #2 on the length of the paper also, we attempt to keep these discussions brief.

*A final point is the quality of all the figures in the manuscript that I found very poor and such that to impede a proper understanding of the manuscript.*

Thank you for this feedback, we have replaced the figures mentioned with new marker-line versions (greyscale) and increased the image resolution to improve the quality.

*Based on that I recommend publication after major revisions. In the following, the authors can find a list of comments with the associated relevance listed in order of appearance in the manuscript.*

**COMMENTS**

*Pag. 2 Lines 53-76 MINOR: the authors may also cite the work of Moramarco et al. (2013) which uses an interesting method for identifying the flow depth distribution in natural channels.*

*Moramarco, T., Corato, G., Melone, F., Singh, V.P., 2013. An entropy-based method for determining the flow depth distribution in natural channels. Journal of Hydrology, 497,176-188.*

This reference has been added to the paper.

*Pag. 3 Lines 88-91 MINOR: Can you rephrase this sentence more clearly?*

Lines 88-91 have been rephrased to:

   *"In particular the methodology uses flood extent with an accuracy-scoring method that disregards the correct detection of 'no water' pixels"*

*Pag. 4 Lines 120-128 MAJOR: If I understand correctly the channel depth is expressed as H=r\*B where H is the channel depth, and B is the width of the channel. Since the hypothesis of linear scaling is central in the study, I thinks this part deserves more profound discussion about: 1) How much it will affect the results of the study. 2) Which are the expected problematics associated with the uniform channel depth.*

Extra discussion has been inserted at the end of the manuscript around the limitations of our assumptions and what affect these might have on the results.

*See also the paper of Yan et al. (2014) where H is a free parameter of the model uniform along the river reach.*

*Yan, Kun, et al. "Exploring the potential of SRTM topography and radar altimetry to support flood propagation modeling: Danube case study." Journal of Hydrologic Engineering 20.2 (2014): 04014048.*

This reference has been added to the paper.

*Pag 3 Section 1.2. MINOR: A scheme or figure of the method would significantly help to understand the image-processing algorithm.*

A good point, the authors have inserted a new figure to illustrate the steps of the methodology: Figure 1.

*Pag. 6 Lines 211-213: MODERATE: it is not clear how the procedure is used with multiple images. Please provide more details.*

A description of the procedure for combining image results has been updated in section 1.4:

   *"These group scores are determined by multiplying each single model/SAR flood map CSI result with the CSI score of the next SAR flood map until all members of the particular group have been added. The unique combinations which comprise these groups are described in Table 3 below. This combining of CSI scores is done for results from each of the 1000 models/parameter*

*scenarios. The next step is the same as for single CSI scores as described above – i.e. to rescale the objective function and compute the cumulative sup*

**Pag 7 Figure 1 MINOR: The quality of this figure is very poor. Please provide a larger and cleared picture where the identification of the study area and the boundary conditions are more clearly visible.**

Thank you for pointing this out, Figure 1 (now Figure 2) has now been amended to be clearer with boundary locations highlighted.

**Pag 11 Figure 2 MODERATE: the quality and the description of this figure is very poor. Also, ENVISAT and Aerial data seem to be a bit different although with this picture is very difficult to compare the results. I understand that the processing of the SAR image inherently contain errors, I am wondering if the results of the paper might be affected by these errors. The authors could test the procedure also on the aerial photograph to understand the effect of the errors in the processing of the image or on the observed stages.**

**The observed model which is expected to behave better than the test model seems to be worse than the test model? Do you have a justification for that? Does this depend on the calibration?**

A well observed point for Figure 2 (now Figure 3) in the paper. The description of this figure in the text has been updated so it is clearer why it was inserted. Also the CSI scores embedded in the figure have been moved to a table for easier comparison/interpretation for the reader. The figures themselves (particularly the modelled and ENVISAT flood extents) have a coarse resolution that does not reproduce very nicely on the page unfortunately.

This particular aerial flood extent image was derived from a single aerial photograph of the flood of July 2007 on the Severn by manual delineation and so is restricted both by the limits of the photographs (cutting off the upper River Avon for example) and interpretation of the image in terms of the flood boundary through vegetation. In this case we used the aerial data here more for validation purposes rather than explicitly to test the calibration methodology. The aim of the paper was to test a series of more 'moderate' resolution imagery that is more extensively and frequently available.

However the authors have added some additional text around the new table of aerial versus ENVISAT CSI scores to explain further the test results we found when applying the methodology on the single aerial flood extent data. A good comparison could be made between the methodology applied to the ENVISAT observation (acquired 23[rd] July 2007, 10:27am) and the subsequent aerial image (acquired 24[th] July 2007, 11:30am) as they were observing the flood merely 24hrs apart. While it is quite likely that the impact of SAR processing errors would be manifest in the final results, and be most obvious when compared with better resolution data such as can be obtained from a gauge record or aerial imagery, the authors felt that an explanation of the impact of these errors was beyond the scope of the paper (which is rather long already).

In Figure 2 (now Figure 3), it is obvious that the observed model (constructed using surveyed cross sections) has not represented observed flood extent as well as the test model. This is most evident in the tributaries to the main River Severn. We updated the text in section 3.1 of the paper to explain this:

*"The scores and flood extent for the observed model are not better than the test model results as might be expected. This may be explained by the fact that while the bathymetry of the observed model does come from survey data, the (domain-average) channel roughness value is not calibrated in either model. While the test model had 1000 parameter-varying depth and roughness values, the observed model had a best estimate of domain-average channel roughness parameter (of 0.038). While appropriate for the main rivers, it is evident that the channel roughness value is not suitable for the narrower tributaries."*

**Pag 11 Figure 3 MAJOR: Please provide a better figure with colors. It is very difficult (with this figure) to follow the authors' statements.**

This contour plot has now been updated in greyscale so it is easier to read with two single opposing colours than the original colour spectrum which made it difficult to see where CSI was highest and lowest, even in full colour. Plus this version should be more printer friendly in black and white.

[Figure]

**From the manuscript: figure 4 : Single SAR acquisitions are compared with LISFLOOD-FP modelled flood maps for the July 2007 flood event. Left: results from the SAR acquisition on 23rd July 2007 at 10:27, right: result from the SAR acquisition 24th January 2008 at 10:12.**

**Pag 11 Section 3.1 MODERATE: I found this section very difficult to follow and to read. I suggest to try to present it better.**

A re-write of this section has been carried out and the authors hope it is now easier to read and follow.

**Pag 12 lines 346-349, MAJOR: the authors concluded that nc is insensitive when estimated simultaneously with the channel depth. However, it seems that this was concluded based only on two images (23rd July 10:27, and 17 January 2008 21:55). Do the authors exclude that this is true in any case and there are not effects of the time of acquisition and the magnitude of the flood event? I think the authors should provide more proofs for this statement. Overall, I find this assumption reasonable however I think that including the DYNIA also for the parameter nc would add a lot of value to the paper.**

This paragraph has been rewritten within the context of the above comment on section 3.1 and hopefully now better explains why '$n_c$' results were excluded from the final version of the paper.   All

available SAR data were analysed for '$r$' and '$n_c$' though only two '$r$' plots featured in the paper for simplicity. As the authors hope has been explained sufficiently within the general comments section above, the '$n_c$' parameter did indeed undergo the same DYNIA analysis as the depth parameter but results were not exceptional enough to be included within an already long paper. The same lack of responsiveness of this parameter in the results were observed for all SAR data analysed, and even when the SAR data were 'grouped' the identifiability and IC results did not greatly improve for '$n_c$' .

With this available dataset of ENVISAT SAR images and hydraulic model set up, the authors have concluded that '$n_c$' is insensitive when estimated simultaneously with the channel depth parameter '$r$'. However additional testing would need to be carried out to conclusively say that this insensitivity is true with other SAR data and magnitudes of flooding.

***Pag. 13 line 361. MODERATE: It is not clear how the IC score is calculated for multiple images.***

The grouping of SAR data occurs after each single model/SAR flood map is assessed and given a CSI score. It is the CSI scores which are multiplied together in particular combinations (as described in Table 3) before the DYNIA methodology is applied and identifiability and IC determined. A group IC score is derived from the grouped SAR data and CSI scores multiplied together. This explanation has been inserted in to the paper in section 3.2 and 1.4 for greater clarity.

***Figures 4, 5, 6, 7 and 8 MAJOR: The interpretation of these figures must be described in the method section. From the text it is very difficult to follow the authors' statements. Please provide a better quality figures as well. It is impossible to discriminate between the different lines. If I understand correctly these are the cumulative distribution of the rescaled support values and not the gradient. The gradient should refer to their slopes. Isn't? If so, I expect a figure like the one in the paper of Wagener et al. (2003), (see FIGURE 8 in their paper).***

Thank you for pointing out how difficult these figures are to read. The original figure coloured lines have been replaced with marker-line figures (greyscale) in this revised version and an explanation of how the cdf plots should be interpreted is now inserted into the method section.

It is correct that Figures 4 – 8 in the paper are representative of the (gradient of the) cumulative distribution of rescaled support values. The gradient plots were converted to cumulative distribution function (cdf) plots, and this is what was shown in the final version of the paper. This was a change introduced to make the plots easier to read together and normalised, when the original histogram plots were numerous and previously side-by-side for each grouping.

For this experiment, the SAR data acquisitions are not regular or plentiful through time (as in the experiment of Wagener e*t al.*, 2003 which used 6 years of continuous daily flow data) and so it was more difficult to represent in a line plot. However for purposes of explanation, the last cdf plot in the paper is converted into a 'gradient' plot for parameter '$r$' below (top figure) to mimic Figure 8 in the paper of Wagener *et al.*, 2003. We have amended the Figures 4-8 (now Figures 5-9) within the paper so they more closely resemble the Wagener identifiability plots.

[Figure]

[Figure]

**Figure 4 top: a new gradient of the cumulative distribution of rescaled support values for each SAR grouping, and bottom: the original cdf of the gradient of the cumulative distribution of rescaled support values for each SAR grouping, as featured in the paper.**

*Pag 18 lines 503. MODERATE: No colors can be seen in the figures.*

Figures have been redone and text updated accordingly.

*Pag 19 lines 545-556. MODERATE: I expect here some discussions about the possible consequences of the assumptions made in the paper.*

This consequences of our assumptions in the paper have been expanded within the results and discussion sections, as described above.

**Reviewer Guy Schumann (Reviewer #2)**

*The IC approach is really nice and gives an objective assessment of the value of a flood image for calibration. However, what we are still missing in the literature is to find a way that gives an objective IC of a SAR flood map without the need to calibrate first. In other words, in this paper, which I think has a lot of merit, IC is built up based on parameter identifiability rather than for instance inter-comparing each SAR image and applying the score and identifiability that way, so without the need of a model and its parameter but I understand that this is outside the scope of this paper.*

It is an appealing idea to attribute IC to individual SAR data, without need of using models and parameters for calibration. At the moment the authors are not sure how to inter-compare SAR data to reveal information content but it could be an interesting topic of further study.

*I also think that what is innovative here is the analysis of IC and identifiability in relation to what stage in the hydrograph we are looking at and what type of data we use (single image, combined images, gauge data). I wonder if the title and the introduction should better reflect that since to me this is one of the first papers to try and answer these questions using real data.*

Thank you for raising this point. The authors have updated the title of the paper and reworded the abstract, so that these reflect more accurately the unique points within the paper.

*My biggest reservation in this study lies with the choice of performance metric used, which may explain in my opinion why the greatest information content is in the SAR images closest to peak flow. Stephens et al. (2014) showed that the performance measure used here is particularly biased towards largest flooded area (in other words, it always gives the highest score to the biggest flooded area). This is significant in this study and could lead to an unwanted "bias" in the calibration. I suggest the authors repeat the exercise offline with the "F2" measure for instance ((A-B)/(A+B+C)) or an area in error index ((B+C)/(A+B+C+D)) to see if the same SAR images give the highest sensitivity still.*

A valid point is raised here. In the preparation for this paper the authors did indeed prepare a number of 'skill score metrics' before deciding on the CSI skill score in preference over 'F' measures and other promising metrics such as Percentage Correct. A sample of these initial plots are shown here in Figure 5 to explain why in the end we decided to use the CSI results:

| Metric | Jul 2007 # 1 at time 10:27 | Jul 2007 #2 at time 21:53 |
|---|---|---|
| **CSI**

 A/(A+B+C) |  |  |
| **F4 / F2**

 (A-B)/(A+B+C) |  |  |
| **PC**

 (A+D)/(A+B+C+D) |  |  |

**Figure 5 Skill Scores using a range of performance metrics. Top: CSI, Middle: F4 or F2 Score, Bottom: Percentage Correct**

These are just a sample of the results available and show the results from analysis of 2 SAR data from the flood of July 2007, but they are fairly representative of all the results for the full range of SAR-derived flood maps which were analysed. The F2 (aka F4) and CSI plots in particular gave a more sensitive response through the changing parameter value '*r*'. This was no doubt due to the 'white space' of no-water cells being absent from the skill score equation. Considering all SAR results together, the CSI scores were observed to provide a more responsive and consistent result with changing parameter value than the other metrics which were assessed.

Concerning the comment regarding CSI skill scoring usually providing a better result for fuller flood extents due to the dominance of 'water' pixels, as pointed out by Stephens *et al.* (2014). This point is well raised but the authors would respond that the CSI skill score is secondary to the shape of the CSI peak itself in this particular case. This identifiability methodology looks at how sensitive the model is for different parameters around this CSI peak and gives little significance to the CSI scores themselves.

To illustrate also the greater sensitivity/information for '*r*' seen in the images when using CSI, the IC scores using PC and F4/F2 as the central metrics are shown in the table below, against the original scores obtained using CSI:

**Table 2 Information Content (IC) for parameter *r*. Top row: single SAR images from July 2007 flood event, and bottom row: the same 2 data, grouped into 'flood event'.**

| IC for | CSI | F2/F4 | PC |
|---|---|---|---|
| Single SAR data (left: SAR1 at time 10:27, right: SAR2 at time 21:53) | 0.165 / 0.188 | 0.066 / 0.102 | 0.102 / 0.101 |
| July 2007 'flood event' | 0.37 | 0.079 | 0.105 |

[revised manuscript text omitted]
.* (2013)  and Chini *et al.* (under review). This method has three steps as illustrated in Figure 1 below. Firstly the probability density function (pdf) of the open water backscatter values in the SAR data is estimated. This requires identification of the bimodal aspect to a histogram of backscatter values so that 'open water' values can be recognized from other backscatter values. A theoretical pdf of water backscatter is then fitted to this histogram using nonlinear regression techniques. The backscatter threshold value ($Th_{seeds}$) where this pdf starts to diverge from the histogram is identified. Then isolating those pixels with backscatter values lower than this threshold produces a preliminary flood map (region growing seeds). The second step is to apply a region growing approach to grow the flooded areas within the preliminary flood map until a tolerance threshold level is reached ($Th_{tolerance}$). For the SAR image this step refines the extent of pixels with an open water value.

In the last step a reference image is used to remove pixels from the flood map that do not change between the flood and non-flood images (Hostache *et al.,* 2012) – i.e. pixels which have 'water surface like' radar responses and could be either bodies of permanent water or smooth surfaces such as car parks or flat roofs. This third step creates the final binary map of flood extent. Errors inherent in the SAR processing are, for simplicity, not considered in this paper.

[Figure]

**Figure 1 - General scheme of the three processing steps of the flood detection algorithm.**

**1.3   Performance measures**

We compare these SAR derived flood maps against the simulated flood maps generated from LISFLOOD-FP output at the equivalent time step by using a contingency matrix shown in Table 1. Flood maps are compared pixel to pixel to determine if there is agreement or disagreement between the two paired maps on whether there is surface water present or not.

**Table 1 - Contingency table (after Stephens et al, 2014 and Mason, 2003).**

| | | Modelled | |
|---|---|---|---|
| | | **Water** | **No Water** |
| **Observed** | **Water** | A)   Correct Water (Hits) | B)   Under-prediction (Misses) |
| | **No Water** | C)   Over-prediction (False Alarms) | D)   Correct No Water (Correct Rejections) |

From this a binary pattern performance measure is used to give a deterministic indication of how well each LISFLOOD-FP simulated flood map has represented the observed data (Mason, 2003 and Stephens *et al.*, 2014). We chose to use the Critical Success Index (CSI, equation 1 below) as this measure does not consider 'correct rejections' ($D$ in Table 1) in the calculation (Bates and De Roo, 2000, Horritt *et al.,* 2001a, Aronica *et al.,* 2002) and it weights over- and under-prediction equally (*C*

and *B* respectively). CSI scales between 1 (indicating perfect skill in the model) and 0 (indicating no skill in the model).

$$CSI = \frac{A}{A+B+C} \tag{1}$$

If 'correct rejections' were included by the use of a different performance measure the result would be overly optimistic scores, given the large areas of 'no water' normally observed in a SAR image. All LISFLOOD-FP simulated flood maps would seem to perform exceptionally well with little to help differentiate between each simulation.

Before comparing SAR and LISFLOOD-FP model results an independent remote dataset is used to illustrate the impact of observation errors and gaps inherent in the SAR data from processing. This validation step makes use of a very high-resolution (0.2m) aerial photograph taken by the EA on 24 July 2007 from an aircraft passing over at 11:30 GMT (details within Giustarini *et al.,* 2013). A flood map shapefile was created from this imagery by manual definition of the flood boundary. This was then converted and upscaled to a raster with the same spatial resolution (75m) of the LISFLOOD-FP model results. Both the ENVISAT data and the LISFLOOD-FP results (the highest scoring models) are compared with this aerial data. A figure showing these flood extents and the CSI results from this comparison are given in section 3.1 below.

**1.4   Parameter identifiability**

[revised manuscript text omitted]

**3 Results and discussion**

**3.1 CSI scores**

In this paper we compare the results of hydraulic model-generated flood maps with the SAR observations of flood extent in order to determine if the satellite data has information in terms of calibrating the model. However with inherent errors in the SAR data from processing it is worthwhile first to compare the SAR data with those from other available remote data to illustrate the impact of observation errors. For validation, the CSI score is calculated between the ENVISAT data and an aerial photograph of the River Severn taken on 24[th] July 2007.

Figure 3 illustrates the derived flood extent from this aerial data (far left) with the ENVISAT WSM SAR derived flood map (centre left) from the previous day. Highest scoring LISFLOOD-FP simulation flood maps from the 'observed' model (centre right) and 'test' model (far right) at the same time step as the ENVISAT data are included for comparison. The CSI results from this SAR-aerial and SAR-LISFLOOD-FP model comparison are shown in Table 4.

**Table 4 - CSI scores, for July 2007 flood extent maps. Comparing results obtained using ENVISAT WSM SAR and aerial derived flood extents, with hydraulic model generated flood extent.**

| Flood map: | Aerial photograph derived | ENVISAT derived |
|---|---|---|
| **Aerial photograph derived** | - | 0.47 |
| **'Observed' Model (not calibrated)** | 0.74 | 0.43 |
| **'Test' Model** | 0.75 | 0.46 |

It is clear that the observed and test LISFLOOD-FP models produce lower CSI scores with the SAR data than with the aerial data. This is to be expected and other studies, which have used higher resolution SAR imagery for validation (e.g. Bates *et al.* 2006, Di Baldassarre *et al*. 2009a and 2010), have observed the same result. The aerial photograph-derived flood map was delineated manually and therefore has improved representation of flooding because there are no detection gaps in the flood extent, whereas SAR-derived flood extents rely on the correct detection of areas of water using a procedure which is vulnerable to issues of detection and processing. So while we may conclude that aerial imagery has the best level of detail in flood extent available here, this data can also be limited by observation extent and processing (i.e. manual delineation of the flood edge is still interpretive) and as a resource is not as frequently available as SAR data for observing flood events. It is also worth pointing out that for the ENVISAT SAR data, describing flood extent using the semi-automated algorithm can be a faster solution than manually delineating flood extent from new photographs.

[Figure]

**Figure 3 - The July 2007 flood extents as observed by aerial photography (on 24rd July 2007 at 11:30 , left) and ENVISAT ASAR instruments in WSM (on 23rd at 10:27, centre left). The same flood event simulated in LISFLOOD-FP with surveyed cross sections (centre right, with Manning's channel roughness fixed at 0.038) and the test model with optimally calibrated parameters (right).**

The scores and flood extent for the observed model are not better than the test model results as might be expected. This may be explained by the fact that while the bathymetry of the observed model does come from survey data, the (domain-average) channel roughness value is not calibrated in either model. While the test model had 1000 parameter-varying depth and roughness values, the observed model had a best estimate of domain-average channel roughness parameter (of 0.038). While appropriate for the main rivers, it is evident that the channel roughness value is not suitable for the narrower tributaries.

Of interest also, when the aerial data is compared with the ENVISAT WSM SAR derived flood maps (row 1, last column), CSI scores are similar to those obtained from the best hydraulic model results. This indicates that the hydraulic models are representing the observed flood extent for this flood accurately, within the limits of the available data. While sections of the flood are missing in the SAR data (for example upper River Avon and Severn) bias can be introduced. Ideally these non-informative areas of the SAR data would be masked out to limit the impact, but with series of data each differently capturing a flood event this requires a more comprehensive analysis than available here. It is currently is an active area of research; for example Giustarini *et al.* (submitted) propose flood probability maps from sequences of SAR data. These maps could be used to mask out 'low probability of flooding' areas. Also Schlaffer *et al.*, 2015 makes use of harmonic analysis to refine flood extent mapping – a mask could be created to obscure pixels with low signal to noise ratios.

As explained in section 1 the first step in the methodology is to examine the accuracy of the test model with changing parameter value, using CSI. The ENVISAT WSM SAR and LISFLOOD-FP CSI results were plotted against the '$r$' and '$n_c$' parameter variables and are presented in Figure 4. This figure includes only two plots: one for an ENVISAT WSM acquisition taken on 23rd July 2007 (10:27am) and one taken on 24th January 2008 (10:12am), but these CSI results represent typical results for the entire SAR data available.

[Figure]

**Figure 4 - Single SAR acquisitions are compared with LISFLOOD-FP modelled flood maps. Left: results from the SAR acquisition on 23rd July 2007 at 10:27, right: result from the SAR acquisition 24th January 2008 at 10:12.**

The black areas in Figure 4 show that a number of '$r$' and '$n_c$' parameter combinations/models are able to produce a good result (i.e. equifinality as described by Beven, 2009). The optimal '$r$' parameter range varies slightly depending on the image considered. Here test models with the best reproduction of the SAR flood map have '$r$' parameters between approximately 0.10 and 0.30 (July 2007) and

between 0.07 and 0.25 (January 2008). Generally, the best reproduction of the SAR flood maps is obtained with models that have an 'r' value in the smaller parameter range which translates to a wide and shallow river channel.

Figure 4 also illustrates the co-variance and a linear dependency between the two parameters. This was observed in all the SAR data. Although the choice of parameter range emphasizes it, there is a slightly greater skill score sensitivity to changes in '$r$' than for '$n_c$'. This is to be expected since changes in channel depth would have an immediate and local impact on flood level and flood extent. It is logical therefore to see changes in '$r$' producing a marked change in flood extent. Channel roughness changes by contrast have an impact more on flow velocities, consequently impacting on the timing of flood wave propagation through the channel (as discussed in Neal *et al.,* 2015). This would have a more spatially diffuse impact on flood extent that is barely perceptible here.

Previous SAR based assimilation studies (Hostache *et al.,* 2009, Mason *et al.* 2009, Di Baldassarre *et al.* 2009a) show that with a known and fixed channel bathymetry there is sufficient sensitivity in the roughness parameter to enable calibration. The above findings indicate that the sensitivity of '$n_c$' is less obvious when '$r$' is also unknown. There are previous studies also where, as here, channel friction appears less sensitive when other parameters are simultaneously calibrated. Roux *et al*. (2008) for example found sensitivity in hydraulic model response to channel roughness to be weaker than sensitivity to geometry parameters and boundary conditions within a Generalised Sensitivity Analysis framework. Additionally Garcia-Pintado *et al.* (2015) found that sensitivity to bathymetry parameters dominated when using the Ensemble Transform Kalman Filter to simultaneously estimate bathymetry and channel friction. The sensitivity in channel friction may therefore be not as obvious when other parameters are simultaneously calibrated because the model is no longer compensating for previously unrepresented uncertainties. It could be suggested that channel friction is reverting to its true sensitivity and so when channel friction is combined with more dominant parameters such as channel bathymetry it is rendered less useful for model calibration.

Consequently an important result of this paper is that - in this particular experimental set up with channel roughness parameter '$n_c$' examined simultaneously with the channel depth parameter '$r$' for the available ENVISAT SAR data - '$n_c$' has a much reduced sensitivity compared with the '$r$' depth parameter response.  It is observed that '$n_c$' will yield optimal results for as long as '$r$' is also unknown. This lack of sensitivity of channel roughness in this and all subsequent results meant that '$n_c$' could not be identified with any real confidence with this methodology (while '$r$' is also unknown). So while '$n_c$' analysis was carried out, from this point onwards only those results from the more identifiable '$r$' parameter are shown. '$n_c$' results are now omitted (but can be provided upon request if of interest).

**3.2   Information content (IC)**

Table 5 presents IC results for depth parameter '$r$'. For single SAR observations (left column) there is clearly greater information content in the July 2007 flood event images. The inundation during this higher magnitude event extended well into the floodplain and the flood detection algorithm was able to detect a large number of flooded cells. The lower IC scores for the March 2007, January 2008 and January 2010 events show that these observations contain less information to help estimate parameter '$r$'.

**Table 5 - Information content for '*r*' from SAR observations and groups of SAR observation with a 90% confidence limit applied.**

| Sequence | Information Content | Sequence | Information Content |
|---|---|---|---|
| 1 - Mar07_1 | 0.10 | Rising limb | 0.13 |
| 2 - Mar07_2 | 0.11 | Peak of hydrograph | 0.23 |
| 3 - Mar07_3 | 0.11 | Falling limb | 0.64 |
| 4 - Mar07_4 | 0.11 | March 07 event | 0.50 |
| 5 - Jul07_1 | 0.16 | July 07 event | 0.37 |
| 6 - Jul07_2 | 0.19 | January 08 event | 0.25 |
| 7 - Jan08_1 | 0.10 | January 10 event | 0.14 |
| 8 - Jan08_2 | 0.11 | All SAR  [1-11] | 0.68 |
| 9 - Jan08_3 | 0.11 | | |
| 10 - Jan10_1 | 0.10 | | |
| 11 - Jan10_2 | 0.10 | | |

Grouping SAR data boosts the IC scores considerably as can be seen in the right hand side columns of Table 5. Group IC scores are estimated after the SAR data have been grouped together and CSI scores combined as described in section 1.4. Different SAR groupings were tested as illustrated in Table 3 including combinations according to flood event, position on the hydrograph as well as 'all SAR' data.

For IC the July 2007 flood now no longer outperforms the rest and instead combinations of images, like the March 2007 flood event, have greater information on '*r*'. The March 2007 flood combination combines observations either side of the hydrograph peak and the January 2008 flood combination observes flooding 'at peak' and soon after in the falling limb. By contrast the reduced-scoring January 2010 and July 2007 combinations acquired images at a single stage in the hydrograph only. We might conclude that the detection quality of the SAR flood maps and timing of acquisition must influence the final IC score and this is supported also by the observation that the early 'falling limb' grouping has one of the largest IC scores here.

Nevertheless, the number of SAR flood maps combined appears to be important also since the 'all SAR' and early 'falling limb' (just over half of these SAR images, Table 3) groupings emerge as providing the highest IC. The March 2007 flood grouping also contains twice as many members as the July 2007 or January 2010 flood groupings and outperforms both. Clearly, incorporating data from multiple observations improves IC since combining SAR images (and CSI scores) improves the likelihood of extracting information on the unknown parameters. However it is not simply a question of numbers otherwise 'falling limb' (combining 6 SAR flood maps for an IC score of 0.64) would not be approaching the success of 'all SAR' (combining 11 SAR flood maps for an IC score of 0.68). Nor is greater information necessarily revealed by removing poor scorers ('all SAR' IC score reduces from 0.68 to 0.64 when the 4 lowest scoring flood maps are removed from this grouping). Instead the solution may lie in using SAR flood maps around the peak and falling limb of the flood since combining 'falling limb' and 'rising limb' observations together yields an IC score of 0.65 but combining 'falling limb' and 'peak' observations together provides an IC score of 0.67. Further work and data is necessary to draw any firm conclusions for the '*r*' model parameter.

**3.3 Identifiability**

The identifiability of '*r*' within single images and combinations of images is assessed in this section. This shows where the parameter is most easily identified in the ensemble of model results. A strong identifiability response would be marked out by having a sharper peak in the following plots. The steeper the gradient, the stronger is the identifiability of the parameter. A sharper peak indicates that the best performing parameters are concentrated in a small area of the parameter space. Conversely a wider, shallower peak would indicate lower identifiability and that the best performing models are widely distributed within the parameter range.

From the CSI contour plots as illustrated in Figure 4 we see that the best performing model parameter combinations are distributed fairly evenly within the parameter space so a 90% confidence limit was also applied to the data prior to measuring the gradient of cumulative distribution of rescaled support values and creation of these following plots.

**3.3.1 Individual SAR observations**

[Figure]

Figure 5 - Identifiability against '*r*' parameter, for each ENVISAT SAR observation in archive.

Figure 5 shows the identifiability plots for all single SAR data, numbered as in Table 2. Because these plots do not generally have a strong peak, identifiability is relatively weak for the individual SAR observations. The strongest response here occurs for '*r*' between 0.05 and 0.15. The peaks are shaped differently for each SAR observation; SAR 4 and SAR 3 both have stronger identifiability (narrower peaks than the rest) whereas SAR 6 and SAR 2 are relatively weak in this ensemble by having wider peaks.

Taken collectively these data provide inconclusive results. This generally weaker identifiability suggests that parameter '*r*' would be difficult to identify within this data individually. The SAR data

were acquired during different flood events (see Table 3) and their peaks occur at different *'r'* parameter values. This variation may be due to differences in the size of flood extent (magnitude of flooding), the processing of the image or simply how the flood developed and that the location of flooded pixels is important.

**3.3.2 Flood event**

This section illustrates identifiability when data from individual SAR images are combined into 'flood events' as indicated in Table 3. An important characteristic of the 'flood event' identifiability plots is that the SAR acquisitions are taken together in close sequence. Garcia-Pintado *et al.* (2013) found that a tight sequence of images could improve model predictions. Combining observations in this way appears to focus the location of the *'r'* parameter more clearly than is possible using single images.

[Figure]

**Figure 6 -Identifiability against parameter *'r'*, for flood events.**

This plot shows that the March 2007 and January 2008 events produce a stronger identifiability, between *'r'* parameter values 0.07 and 0.15. However, the optimum *'r'* value varies between 0.07 to 0.1 and 0.1 to 0.15 depending on which of these floods is examined. It is entirely reasonable that identifiability of channel depth parameter in the data would vary with flood event as each flood is unique in magnitude and mechanism. Based on Figure 6, the March 2007 and January 2008 SAR images might therefore be best utilised to locate the value of parameter *'r'*. These two events have approximately the same peak discharge flows at Saxons Lode (see Table 2). However, the IC results point towards the March 2007 data combination alone as having more parameter information and the reason for this becomes clear when looking at the individual SAR maps of flood extent. The group of SAR images acquired in March 2007 combine to yield a more complete representation of the flood development than the combination from January 2008. So although Figure 6this identifiability plot shows that both March 2007 and January 2008 flood events would be useful to locate the parameter *'r'*, IC shows the information contained in the March 2007 flood maps to be of most value.

**3.3.3 Through the flood hydrograph**

[revised manuscript text omitted]

Figure 9 compares the identifiability for 'All SAR' data for the full range of models (roughness is not constrained, solid line) and for 236 models which satisfy the constraint of having '$n_c$' between 0.035 and 0.055 (dashed line). Where there is no constraint on '$n_c$' the location of '$r$' is most identifiable between approximately 0.10 and 0.12 in 'All SAR' groupings. With '$n_c$' constrained the '$r$' value moves to a lower depth range of between approximately 0.08 and 0.10. This translates to a reach-average model depth of between 6m and 7.2m and is reasonably close to the observed data. In this constrained group of models, the single highest scoring model has '$r$' of 0.086 ('$n_c$' of 0.036) and so indicating the optimum reach-average model depth is around 6.51m. The equivalent rectangular depth from the EA survey is 5.63m (assuming a reach median width of 76m) using bank-full cross sectional area. The difference therefore between the calibrated value and the observed equivalent is approximately 0.88m (an error of 16%).

The model responds to changes in channel friction by altering the speed of the flood wave and flow velocities. These results highlight the important reasons for calibrating this second parameter concurrently. If channel roughness were set too high the flood wave would be delayed. Set too low and the flood wave would be too advanced.

**4   Conclusion**

This paper presents a methodology for dual calibration of bankfull depth and channel roughness parameters of the LISFLOOD-FP Sub-Grid hydraulic model using SAR data and a binary pattern classification measure based on flood extent. Multiple models performed well initially, but by employing an identifiability methodology we located the area of the parameter space with highest information for the depth parameter '$r$'. The location narrows with the use of more SAR images.

The methodology provides some information on which single and combinations of SAR flood maps would be most useful for calibration purposes. Single SAR flood maps would be sufficient to calibrate the depth parameter but the identifiability is much improved when multiple maps are combined. Combinations aligned according to particular flood events/magnitudes are not conclusively different, but using many or all available SAR images does offer a real improvement in identifiability. There are indications that combining maps with similar flood duration, or stage of flood (i.e. SAR images acquired close to peak or just after) would be beneficial for calibrating the reach-average depth parameter, but further work is needed with more targeted observations than the 11 used here. For robustness, a good range of flood magnitudes should be used for calibration.

The channel roughness parameter '$n_c$' was less sensitive to variations in flood extent and we failed to locate a representative value for this parameter when '$r$' was also varied. The likely cause probably due to the initial range selected being too broad and the suggestion that depth/bathymetry is the more dominant parameter in the model which largely overrides, at this model scale at least, the significance of channel friction. By constraining '$n_c$' to a more plausible range it was possible to improve the calibration method and further improve the global estimate for the depth parameter. Under this constraint the models with top CSI and identifiability results show that the reach-averaged depth parameter is calibrated to 0.086, translating roughly to a reach-average depth of approximately 6.51m. This is an error of 0.88m compared with an equivalent measure from observed cross section data, where channel depth is approximated as 5.63m.

A benefit of this methodology is that although we used gauged inflows within the model, in theory the calibration methodology should work also with no recourse to ground data if good inflows can be simulated and a good DEM is available. The method also does not require a step to obtain water levels from the flood data. It does however make some simplifications and assumptions. First, the method assumes that as there are no errors in the return signals or processing of the ENVISAT WSM images and the derived flood maps therefore represent the true and full flood extent, however in reality all data have some error. and this would likely impact on the identifiability and IC results here. This is particularly true for single SAR data which are compared against each other but perhaps less easily isolated in grouped SAR data as the combining of data smooths out errors and by accumulation compensates for perceived detection errors in the remotely sensed data. Understanding the impact of these individual errors on the final result would be an interesting follow-on experiment. Neither has the importance of the SAR resolution been tested here.

There is also error likely in the assumptions behind the model set up. For example, we assume that the channel depth can be approximated with a parameter '$r$' which is the ratio between channel depth to width at bankfull flow (i.e. '$r$' is a linear scaling so as width varies, so directly does depth in order to conserve water volumes). There is also the assumption that there is no rate of change between width and depth, so in essence depth and width do not vary along the modelled reach and are therefore uniform within the domain. This fixes '$r$', width and depth to a single value per model, which is applied throughout the domain. This assumption cannot truly represent the reality of channel bankfull flows at particular points in the model, so it can only be used if there is an assumption that results represent a 'reach-average' depth value for the entire modelled domain, based on a reach-average width. In this way, local variations in width, depth and flow can be smoothed out. Straight uniform channels are observed in natural systems only for short stretches of river and so the methodology may be more appropriate within smaller sub-reaches (i.e. 'sub-regions' or tributaries) where hydraulics and

hydrology are similar. or within medium sized catchments with ostensibly negligible variation in domain channel width. Future work will investigate the applicability of the methodology under these conditions.

**5   Acknowledgements**

We thank the Environment Agency of England and Wales for providing the river cross-section data, DEM and gauging station data. The authors would also like to thank the editor and two reviewers for their valuable comments which helped us to improve the manuscript

[revised manuscript text omitted]

Yan, K., Tarpanelli, A., Balint, G., Moramarco, T. and Baldassarre, G.D., 2014. Exploring the potential of SRTM topography and radar altimetry to support flood propagation modeling: Danube case study. Journal of Hydrologic Engineering, 20(2), p.04014048.

Yoon Y., Durand M., Merry C. J., Clark,E. A., Andreadis K. M., & Alsdorf D. E. 2012. *Estimating river bathymetry from data assimilation of synthetic SWOT measurements*. Journal of hydrology, **464.** 363-375.